# Visual Fourier Prompt Tuning

Runjia Zeng[1][*], Cheng Han[2][*], Qifan Wang[3], Chunshu Wu[4], Tong Geng[4],
Lifu Huang[5], Ying Nian Wu[6] and Dongfang Liu[1][†]

[1]Rochester Institute of Technology   [2]University of Missouri - Kansas City
[3]Meta AI   [4]University of Rochester
[5]UC Davis   [6]University of California, Los Angeles

## Abstract

With the scale of Transformer-based vision models continuing to grow, finetuning these large-scale pretrained models for new tasks has become increasingly parameter-intensive. Visual prompt tuning is introduced as a parameter-efficient finetuning (PEFT) method to this trend. Despite its successes, a notable research challenge persists within almost all PEFT approaches: significant performance degradation is observed when there is a substantial disparity between the datasets used in pretraining and finetuning phases. To address this challenge, we draw inspiration from human visual cognition, and propose the Visual Fourier Prompt Tuning (VFPT) method as an effective and efficient solution for adapting large-scale Transformer-based models. Our approach innovatively incorporates the Fast Fourier Transform into prompt embeddings, seamlessly integrating both spatial and frequency domain information. Apart from its inherent simplicity and intuitiveness, VFPT exhibits superior performance across various tasks, offering a general solution to address the data disparity challenge. Empirical results demonstrate that our approach outperforms several state-of-the-art baselines on two benchmarks, with low parameter usage (*e.g.*, 0.57% of model parameters on VTAB-1k) and notable performance enhancements (*e.g.*, 73.20% of mean accuracy on VTAB-1k). Our code is avaliable at `https://github.com/runtsang/VFPT`.

## 1 Introduction

> *"Fourier's theorem is not only one of the most beautiful results of modern analysis, but it may be said to furnish an indispensable instrument in the treatment of nearly every recondite question in modern physics."*
>
> − Lord William Thomson Kelvin [1]

Prompt tuning [2, 3] is initially introduced for parameter-efficient adaptation of large foundation models in natural language processing (NLP). As vision models continue to scale for enhanced performance, visual prompt tuning [4] has been applied to various vision domains (*e.g.*, image classification [5], segmentation [6, 7], detection [8]), demonstrating superior performance and lower parameter usage compared to other parameter-efficient fine-tuning (PEFT) methods. However, a common challenge within the research community remains unaddressed: significant performance degradation occurs when there is a substantial disparity between the data used in pretraining and finetuning [9, 10]. This issue hinders the broader application of visual prompt tuning. Consequently, a natural question arises: ① *Can prompt tuning generalize across datasets with varying disparities?*

As researchers commonly draw insights from human to replicate the principles in intelligent machines [11, 12, 13, 14], we consider to answer this question from the human visual cognition's perspective. While humans comprehend the world through past experiences/knowledge, it is essential to generalize and adapt this understanding to new tasks efficiently and effectively. The robust and rapid adaptability of human visual cognition thus arises from various domain analysis, capturing the new patterns from different channels and perspectives [15, 16, 17].

---

[*] Equal contribution.   [†] Corresponding author.

38th Conference on Neural Information Processing Systems (NeurIPS 2024).

Interestingly, we find that the paradigm of visual prompt tuning is conceptually analogous to human visual cognition. While the frozen large-scale vision model functions as accumulated knowledge, the fast adaptation mechanism resembles visual prompt tuning, requiring the incorporation of diverse domains of information (*e.g.*, time, frequency) to achieve comprehensive understandings [18, 19, 20]. The Fast Fourier Transform (FFT) [18, 19, 20], renowned for its ability to convert signals from their original domain (*e.g.*, time or spatial) to the frequency domain and vice versa, serves as an ideal tool for contributing informative insights in the frequency domain. By leveraging the capabilities of FFT, visual prompts can naturally integrate both spatial and frequency domain information during finetuning, thereby enabling the frozen vision model to achieve consistent and robust performance across datasets with varying disparities. Consequently, our research question evolves into: ② *How can FFT be integrated into visual prompt tuning to emulate the human visual mechanism?*

To this end, we employ a simple yet effective strategy that utilizes the Fourier operations to facilitate visual prompt tuning (see Fig. 1(c)). By integrating frequency domain information into learnable prompt embeddings, our approach elegantly assimilates data from both spatial and frequency domains, simulating the human visual cognition. We name our approach **Visual Fourier Prompt Tuning** (**VFPT**), which exhibits several compelling advantages: ❶ *Simplicity.* The intuitive application of FFT in prompt tuning emulates the rapid processing capabilities of the human visual system, making VFPT both elegant and straightforward to implement (see §2.1). ❷ *Generality.* By incorporating frequency domain information, the search space for latent embeddings of prompts is naturally expanded, resulting in advanced enhancement in performance across different datasets and tasks with varying data disparities (see §4.2). The generality of our model is further illustrated through our analysis of the optimization process, which enables smoother navigation towards local minima, increasing flatness around them and exhibiting apparent convexity. ❸ *Interpretability.* To intuitively demonstrate the advantages of Fourier components, we visually illustrate that the introduction of Fourier transform in visual prompt tuning results in a markedly higher concentration of attention scores within the Transformer's input space, which correlates positively with enhancements in performance (see §4.4). This observation, in turn, explains the effectiveness of our approach.

Comprehensive experiments are conducted to evaluate the performance of VFPT. In §2, we conduct a literature review and discuss relevant works. Our approach is presented in §3, where we describe how we simple yet effectively integrate FFT into visual prompt tuning. In §4.2, we present compelling experimental results on various benchmarks, backbones, and different pretraining objectives, achieving superior performance *without* complex engineering design. Specifically, our approach achieves an average improvement of **7.63%** in accuracy on VTAB-1k compared to full finetuning, and **3.77%** compared to VPT [4]. In §4.4, we demonstrate that the FFT prompts significantly enhance the activation of the frozen vision model. Additionally, we study the optimization process of prompt tuning approaches, indicating that VFPT provides a more favorable optimization process. Finally, we demonstrate the strong algorithmic generalization of our approach to the language domain, and show additional visual explanations in the Appendix. We trust that this work provides valuable insights.

## 2 Related Work

### 2.1 Visual Parameter-efficient Finetuning

With the significant growth in the scale of vision models, especially following the emergence of Vision Transformers [21, 22, 23, 24, 25], the development of PEFT methods under "pretrain-then-finetune" paradigm becomes increasingly critical. Current methods under this paradigm can be generally categorized into *partial tuning* [26, 27, 28], *extra module* (*i.e.*, including reparameterization approaches such as Low-Rank Adaptation (LoRA) [29]) [30, 31, 32, 33, 34, 10, 35, 36], and *prompt tuning* [4, 37, 38, 39, 40, 41]. Partial tuning and extra module face several limitations that hinder their application. ① Unsatisfactory performance: they generally cannot reach competitive performance with regard to full finetuning [4, 26, 27, 28, 33, 10]; ② Model-oriented design: most research requires to insert specific architecture/block design [31, 30, 32] during tuning, rendering them non-universal solutions when considering different backbones. In contrast, prompt tuning [2], originally proposed for language-domain [42, 43, 44, 45], provides a general and straightforward solution in vision with powerful performance gains. It signals a new paradigm in PEFT in the field of computer vision.

Generally, prompt tuning introduces a sets of learnable parameters to the input sequence of backbone models, updating only these parameters during the finetuning. Despite its apparent simplicity, the paradigm of visual prompt tuning has demonstrated notable performance enhancements. Current

developments on visual prompt tuning primarily concentrate on engineering optimizations, such as reducing parameter usage [5] and expanding applicability across diverse tasks [39, 46, 47, 48]. These approaches often involve introducing additional constraints and functionalities to the foundational design, which deviate from the principles of simplicity and elegance to the original concept of visual prompt tuning. Our approach, in sharp contrast, endeavors to explore visual prompt tuning from the perspective of ***human visual intelligence***, while diligently maintaining the ***simplicity*** of prompt tuning. It is also essential to emphasize that visual prompt tuning diverges markedly from visual instruction tuning [49] (*i.e.*, aiming at improving the model's instruction following abilities).

## 2.2 Fast Fourier Transform in Vision

FFT is a powerful mathematical algorithm used to compute the Discrete Fourier Transform (DFT) and its inverse [50, 51]. It is pivotal in information processing, allowing the detailed analysis of various signals (*e.g.*, image [52, 53, 54], radar [55, 56, 57]) for frequency determinations. In vision, FFT's ability to transform complex data in spatial domain into frequency domain makes it an invaluable tool for abstracting critical features from noisy or high-dimensional datasets [58, 59]. This abstraction is particularly beneficial as the identification of salient features are shown to have better generalization ability across domains [60, 61, 62, 63], directly influences the performance [64, 65, 66, 67] of image analysis and processing tasks. Current research on FFT in vision predominantly explores areas such as conventional image processing [52, 68, 69, 70], image pre-processing for deep neural networks (DNNs) [71, 72] and DNN architectural design [20, 66, 65, 73, 74, 75, 76].

Despite its profound utility and effectiveness, the integration of FFT within the paradigm of visual prompt tuning remains largely underexplored. Recent work [77] adapts the pretrained multi-modal network to the tasks under modality-incomplete segmentation scenarios via FFT prompt tuning. This approach demonstrates the potential of FFT operations to handle missing modalities (*i.e.*, substantial disparity) effectively. However, it primarily focuses on task-specific optimization and design. The extensive applicability and generality of FFT, especially in cross-dataset analysis, have yet to be recognized or exploited. Another work [36] incorporates Fourier transform into the LoRA-based approach. While the expressive Fourier basis facilitates the recovery of weight changes, it does not fully integrate frequency domain information during finetuning, which remains orthogonal to our approach. In this paper, we aim to broaden the scope of exploration and contribute to advancing the field of Fourier-based research in vision. By studying the integration of FFT with visual prompt tuning, we fully explore how to improve both the efficacy (see §3) and the adaptability of learning models to diverse and challenging datasets (see §4). Furthermore, we present novel evidence indicating that VFPT establishes strong correlations within the Transformer's input space, aligning with the performance enhancements (see §4.4). Overall, the generality of VFPT suggests a novel understanding of the Fourier-based method in current machine learning applications.

# 3 Methodology

In this section, we introduce VFPT, a novel visual prompt tuning approach for effective and general large-scale transformer-based model finetuning. We first define the problem and notations of visual prompt tuning and FFT in §3.1. The integration of Fourier-based visual prompt tuning is presented in §3.2. The overall framework is shown in Fig. 1(c), where we compare our model with original VPT.

## 3.1 Preliminary

**Visual Prompt Tuning.** Given a pretrained Transformer model $\mathbf{T}$ with $N$ layers, the objective of prompt tuning in vision is to finetune a model $\hat{\mathbf{T}}$ into a new task with only a few set of $d$-dimensional embedding vectors, *i.e.*, prompts, in the input space after patch $\mathrm{Emb}$ layer. These learnable prompts are defined as $\mathbf{P} = \{P^1, P^2, \ldots, P^N\}$, where $P^i$ represents the learnable visual prompts in the $i_{th}$ encoder layer. Formally, the encoder layers with prompts are defined as:

$$
\begin{aligned}
Z^1 &= L_1(P^1,\ E) \\
Z^i &= L_i(P^i,\ Z^{i-1}) \quad i = 2, 3, \ldots, N
\end{aligned}
\tag{1}
$$

where the embeddings of the input image patches $E$ are initialized with frozen $\mathrm{Emb}$ projection, and $Z^i$ is the contextual embeddings computed by the $i_{th}$ encoder layer. The colors ■ and ■ indicate trainable and frozen parameters, respectively. Here, trainable prompts only accounts for a small proportion of the total parameters (*e.g.*, 1.14% on VTAB-1k [78] in VPT [4]).

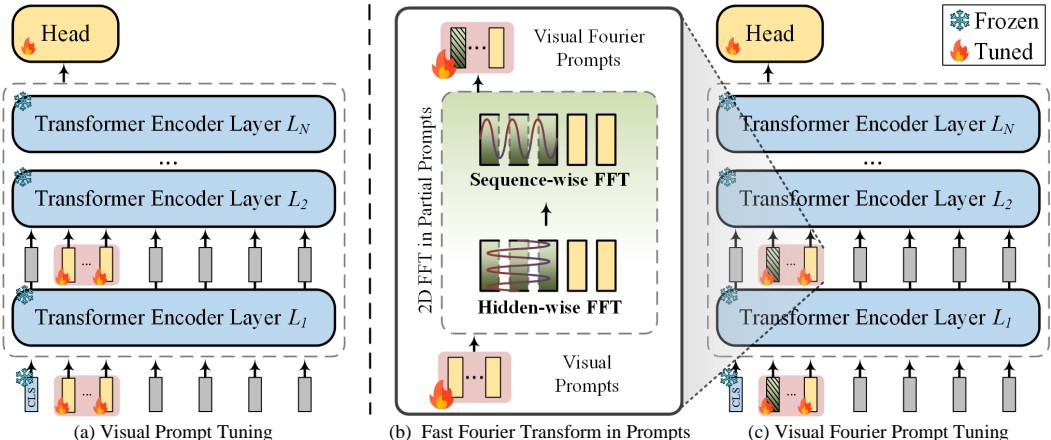

Figure 1: **Overview of VPT** *vs.* **VFPT (ours) frameworks.** (a) Original Visual Prompt Tuning. (b) 2D Fast Fourier Transform operations in partial visual prompts along hidden and sequence length dimensions. (c) The overall architecture of our proposed VFPT (see §3.2).

**Fast Fourier Transform.** The FFT is a powerful algorithm for computing the Discrete Fourier Transform (DFT), which transforms a finite sequence of equally-spaced function samples into a same-length discrete-time Fourier transform sequence. Specifically, given a sequence $\{x_n\}$ where $n$ is a member of the interval $n \in [0, N-1]$, the DFT is defined as:

$$\mathcal{F}(x) = X_k = \sum_{n=0}^{N-1} x_n e^{-i2\pi \frac{k}{N} n}, \quad 0 \le k \le N-1. \tag{2}$$

For a finite sequence of equally-spaced samples $\{x_n\}$, the DFT generates a same-length sequence of equally-spaced samples $\{X_k\}$. This transform is denoted as $\mathcal{F}$. The initial DFT is in complexity $O(n^2)$. For acceleration, we use Cooley–Tukey FFT algorithm [79] following common practice [80] (*i.e.*, complexity $O(n \log n)$). FFT serves as a powerful tool for domain transition. Consequently, we explore the integration of the FFT operation within PEFT methods, particularly in prompt tuning.

## 3.2   Visual Fourier Prompt Tuning

Visual prompt tuning is particularly useful under the *pretrain-then-finetune* paradigm. However, it suffers a significant performance reduction when substantial disparities exist between pretrain and finetune datasets. The reason is that during finetuning on new data, the image distribution may deviate markedly from the examples used in pretraining the backbone model [9]. Existing prompt tuning [4, 5], focusing predominantly on spatial information, can only harness the shared information embedded within the pretrained backbone, limiting their capacity to adapt effectively to novel tasks. Thus, it is crucial to strengthen the ability to capture distinguishing feature from finetuning data.

To this end, we introduce VFPT, an intuitive yet powerful method with advanced performance and generality. Compared to VPT (see Fig. 1(a)), our model (see Fig. 1(c)) transforms partial prompts from spatial domain to frequency domain via 2D FFT (see §3.1) to consider both the spatial and frequency domain information. Formally, for each learnable visual prompts in the $i_{th}$ encoder layer $P^i \in \mathbf{P} = \{P^1, P^2, \dots, P^N\}$, we have $P^i = \{p_1^i, p_2^i, \dots, p_M^i\}$. We select $m$ partial prompts as visual Fourier prompts at each layer, where $0 \le m \le M$. Further, $\alpha = m/M$ represents the fraction of Fourier participation, where zero indicates all prompts are original visual prompts, and one implies all prompts are given after FFT. We apply a 2D FFT on $\alpha$ visual prompt embedding input with respect to both sequence (*i.e.*, $\mathcal{F}_{\text{seq}}$) and hidden dimensions (*i.e.*, $\mathcal{F}_{\text{h}}$). Note that the operations $\mathcal{F}_{\text{seq}}(\mathcal{F}_{\text{h}}(x))$ and $\mathcal{F}_{\text{h}}(\mathcal{F}_{\text{seq}}(x))$ are mathematically equivalent due to the commutative property of the two one-dimensional FFTs [80]. Here, ■ indicates Fourier operations.

$$P_{\mathcal{F}}^i = \Re\left(\mathcal{F}_{\text{seq}}\left(\mathcal{F}_{\text{h}}([p_1^i, p_2^i, \dots, p_m^i])\right)\right). \tag{3}$$

To maintain the pretrained structure's consistency, we only alter the prompt embeddings, and thus retain only the real component (*i.e.*, $\Re$) from the output. This design does not require any adjustments to accommodate complex numbers in the self-attention module, ensuring that the remaining elements

of the model remain unchanged. Consequently, the overall integrated prompts $\hat{P}^i$ in the $i_{th}$ encoder layer are formed by the concatenation between the visual Fourier prompts and visual prompts as:

$$\hat{P}^i = \left[ P_{\mathcal{F}}^i, p_{m+1}^i, \ldots, p_M^i \right]. \tag{4}$$

Our elegant design of VFPT enjoys a few appealing characteristics:

- *Simplicity:* VFPT only requires several lines of code based on the implementation of the visual prompt tuning. Its intuitive integration of information between spatial and frequency domains brings *nearly free* performance efficacy. The low complexity of FFT (*i.e.*, $O(n \log n)$) leads to an overall marginal reduction during the training schedule.(*i.e.*, 2.8% on VTAB-1k [78]). In sharp contrast, current endeavors in visual prompt tuning mainly emphasize augmenting architectural complexity for superior performance [5, 81, 42], undermining the inherent simplicity of prompt tuning and introducing significant training overhead (*e.g.*, [81] learns 2D prompt token map for densely image relationship construction, [5] incorporates additional self-attention K-V prompts).
- *Generality:* The frequency and spatial analysis of imagery inputs can be mutually complementary, leading to a more comprehensive feature understanding from distinct perspectives (*e.g.*, the frequency domain allows for the distraction and decomposition of luminance and noise to a considerable degree [82], while the spatial domain excels in capturing intricate object details). By incorporating learnable prompts from both domains, VFPT demonstrates enhanced prompt learning capabilities, which makes it superior to finetune across diverse tasks (see §4.2). The empirical findings of flatness and convexity of VFPT further strength our claim.
- *Interpretability:* In visual prompt tuning, a notable challenge arises concerning the interpretability of learnable prompts. Unlike in NLP, where tokens explicitly represent these prompts, visual prompts have historically lacked a clear and explainable representation. In order to intuitively perceive the function of visual prompts, we offer a possible way to understand why prompts play an important role in fine-tuning a new task through the visualization of attention maps. Moreover, we can also observe a better and stronger global feature learning pattern through introducing visual Fourier prompts, showing how Fourier prompts work. More discussion will be elaborated in §4.4.

## 4 Experiment

### 4.1 Experiment Setup

**Datasets.** Following common practice [5, 4, 81, 83], our experiments are carried out on two image classification benchmarks. **VTAB-1k** [78] collects 19 benchmarked Visual Task Adaptation, separated into three groups: (1) *Natural* includes natural images captured by standard cameras, (2) *Specialized* consists of images taken by specialized equipment, and (3) *Structured* considers tasks considering geometric comprehension (*i.e.*, counting, distance), which has substantial dataset disparities (*i.e.*, tasks in *Natural* and *Specialized* are closely related to image classification and thus have low disparities, while tasks in *Structured* are regarded as distinct from image classification) when comparing to the pretrained dataset [9] (*i.e.*, ImageNet21K [84]). Each task of VTAB-1k contains 1000 training examples with the $800/200$ split for `train`/`val` set. **FGVC** contains 5 benchmarked Fine-Grained Visual Classification, including CUB-200-2011 [85], NABirds [86], Oxford Flowers [87], Stanford Dogs [88] and Stanford Cars [89]. The training set is split into 90% `train` and 10% `val`.

**Baselines.** For consistency, we follow [4, 5] and compare VFPT with other widely applied parameter-efficient fine-tuning methods. Results of two vision transformer architectures, Vision transformer [23] (ViT) and Swin transformer [24] (Swin), on image classification are discussed in §4.2. We also apply VFPT on two self-supervised objectives: MAE [90] and MoCo v3 [26].

**Training.** Following [4, 5], we conduct grid search to find the best tuning hyperparameters, learning rate (*i.e.*, [50, 25, 10, 5, 2.5, 1, 0.5, 0.25, 0.1, 0.05]), and weight decay (*i.e.*, [0.01, 0.001, 0.0001, 0.0]) on val set. Notably, VFPT ***does not require*** specific-designed large learning rate in [4]. The learning rate is scheduled by a cosine decay policy and trained for 100 epochs.

**Reproducibility.** VFPT is implemented in Pytorch [91]. Experiments are conducted on NVIDIA A100-40GB GPUs. To guarantee reproducibility, our full implementation will be publicly released.

### 4.2 Main Results

In this section, we demonstrate the effectiveness of VFPT from two key perspectives: ♠ *Superior Performance:* Our model demonstrates significant performance improvements across diverse datasets, including challenging tasks with large disparities in data, thus showcasing its generalizability.

Table 1: **Image classification accuracy for ViT-Base/16 [23]** pretrained on supervised ImageNet-21k. Following [4, 5], we report the average test accuracy (three runs) on FGVC [4] and VTAB-1k [78] benchmarks, and "Number of Wins" in [·] compared to full fine-tuning (Full) [92]. ► denotes the method with highest "Number of Wins" compared to Full. We further report "Number of Wins to VPT" in {·}. "Tuned/Total" is the average percentage of tuned parameters required by 24 tasks. "Scope" indicates the tuning scope of each method. "Additional parameters" is the existence of parameters in addition to the pretrained backbone and linear head. **Bold** and **Underline** indicate the best and the second best results. VFPT outperforms full fine-tuning in **22 of 24** instances with fewer trainable parameters and beats VPT in **23 of 24** cases with lower parameters. † denotes methods using soft filtered prompts to reduce the parameter usage in learnable visual prompts, requiring specialized devices to facilitate acceleration. Per-task results are available in Appendix. Same for Table 2 and 3.

| ViT-Base/16 [23] (85.8M) | Tuned/ Total | Scope Input | Scope Backbone | Extra params | FGVC [4] [5] | VTAB-1k [78] [19] Natural [7] | Specialized [4] | Structured [8] | Mean Total |
|---|---|---|---|---|---|---|---|---|---|
| Full [CVPR22][92] | 100.00% | | ✓ | | 88.54% | 75.88% | 83.36% | 47.64% | 65.57% |
| Linear [CVPR22][92] | 0.08% | | | | 79.32% [0] | 68.93% [1] | 77.16% [1] | 26.84% [0] | 52.94% |
| Partial-1 [NeurIPS14][93] | 8.34% | | | | 82.63% [0] | 69.44% [2] | 78.53% [0] | 34.17% [0] | 56.52% |
| MLP-3 [CVPR20][94] | 1.44% | | | ✓ | 79.80% [0] | 67.80% [2] | 72.83% [0] | 30.62% [0] | 53.21% |
| Sidetune [ECCV20][31] | 10.08% | | ✓ | ✓ | 78.35% [0] | 58.21% [0] | 68.12% [0] | 23.41% [0] | 45.65% |
| Bias [NeurIPS17][30] | 0.80% | | ✓ | | 88.41% [3] | 73.30% [3] | 78.25% [0] | 44.09% [2] | 62.05% |
| Adapter [NeurIPS20][32] | 1.02% | | ✓ | ✓ | 85.46% [1] | 70.67% [4] | 77.80% [0] | 33.09% [0] | 62.41% |
| LoRA [ICLR22][35] | — | | ✓ | ✓ | 89.46% [3] | 78.26% [5] | 83.78% [2] | 56.20% [7] | 72.25% |
| AdaptFormer [NeurIPS22][95] | — | | ✓ | ✓ | — | 80.56% [6] | 84.88% [4] | 58.83% [7] | 72.32% |
| ARC_att [NeurIPS23][96] | — | | ✓ | ✓ | 89.12% [4] | 80.41% [7] | **85.55%** [3] | 58.38% [8] | 72.32% |
| VPT-S [ECCV22][4] | 0.16% | ✓ | | ✓ | 84.62% [1] | 76.81% [4] | 79.66% [0] | 46.98% [4] | 64.85% |
| VPT-D [ECCV22][4] | 0.73% | ✓ | | ✓ | 89.11% [6] | 78.48% [6] | 82.43% [2] | 54.98% [8] | 69.43% |
| EXPRES [CVPR23][97] | — | ✓ | | ✓ | — | 79.69% [6] | 84.03% [3] | 54.99% [8] | 70.20% |
| † E2VPT [ICCV23][5] | 0.39% | ✓ | ✓ | ✓ | 89.22% [4] | 80.01% [6] | 84.43% [3] | 57.39% [8] | 71.42% |
| ► Ours | 0.66% | ✓ | | ✓ | **89.24%** [4] {4} | **81.35%** [6] {7} | 84.93% [4] {4} | **60.19%** [8] {8} | **73.20%** |

♡ *Fourier Contribution:* We observe that Fourier components play a critical role in VFPT, where tasks with larger data disparities tend to favor higher percentages of Fourier components.

*Definition of disparity.* Following [9], we use the Fréchet Inception Distance (FID) [99, 100] to measure the disparity between the datasets used in pretraining (*i.e.*, ImageNet) and funetuning (*i.e.*, downstream tasks). Average FID scores of each group are reported in Fig. 2, where the *Natural* group has low disparities due to its close relationship to ImageNet21K [84] and the *Specialized* and *Structured* groups

Table 2: **Image classification accuracy for Swin-Base [24]** pretrained on supervised ImageNet-21k.

| Swin-Base [24] (86.7M) | Tuned/ Total | VTAB-1k [78] [19] Natural [7] | Specialized [4] | Structured [8] |
|---|---|---|---|---|
| Full [ICLR23][98] | 100.00% | 79.10% | 86.21% | 59.65% |
| Linear [ICLR23][98] | 0.06% | 73.52% [5] | 80.77% [0] | 33.52% [0] |
| Partial-1 [NeurIPS14][93] | 14.58% | 73.11% [4] | 81.70% [0] | 34.96% [0] |
| MLP-3 [CVPR20][94] | 2.42% | 73.56% [5] | 75.21% [0] | 35.69% [0] |
| Bias [NeurIPS17][30] | 0.29% | 74.19% [2] | 80.14% [0] | 42.42% [0] |
| VPT [ECCV22][4] | 0.25% | 76.78% [6] | 83.33% [0] | 51.85% [0] |
| † E2VPT [ICCV23][5] | 0.21% | 83.31% [6] | 84.95% [2] | 57.35% [3] |
| ► Ours | 0.27% | **84.53%** [7] {5} | **86.15%** [2] {4} | **58.21%** [3] {6} |

(*i.e.*, orientation prediction task) are considered distinct from image classification. The dataset description of VTAB-1k is covered in §4.1 (FGVC is excluded due to lack of categorization).

♠ *Superior Performance.* In order to have a comprehensive understanding on generality, we examine VFPT on ViT-Base/16 [23], Swin-Base [24], and two self-supervised objectives, following common practice [4, 5]. We also report the individual per-task results for Table 1, 2 and 3 in Appendix.

**VFPT on ViT.** We report the average accuracy score on VTAB-1k and FGVC benchmarks across four diverse task groups for three runs in Table 1, where fifteen protocols under *pretrain-then-finetune* paradigm are considered. Specifically, Full [92] updates both backbone and classification head; Linear [92], Parital-1 [93] (top layer), and MLP-3 [94] (3 MLP layers) are partial tuning approaches; Sidetune [31], Bias [30], Adapter [32], LoRA [35], AdaptFormer [95] and $ARC_{att}$ [96] are extra module methods which add new trainable parameters to backbone for adaptation; VPT-S [4], VPT-D [4], EXPRES [97] and $E^2VPT$ [5] are concurrent visual prompt tuning approaches. Consequently, we have several key observations. *First*, VFPT is able to outperform the full fine-tuning method in **22 out of 24** tasks. For example, our model achieves **0.13%** improvement on FGVC and **5.21%** improvements on VTAB-1k *Structured*, respectively. The empirical results show the effectiveness of VFPT. *Second*, VFPT tunes only **0.66%** of the overall parameters in the backbone, establishing it as a competitive method within the PEFT approaches. *Third*, while VPT struggles to capture the image information when having significant dataset disparity, VFPT achieves notable performance improvements by integrating both spatial and frequency information (see §3.2) without additional architectural modifications. (*i.e.*, **60.19%** *vs.* 54.98% on VTAB-1k *Structured*).

**VFPT on Hierarchical Transformer.** We further extend VFPT to a hierarchical transformer — Swin-Base [24] for architectural generalization. The MSA layer of Swin is employed in local shifted windows, and patch embeddings are merged at deeper layers. For consistency, we follow the same settings from ViT to apply and prepend Fourier prompts ahead of the visual prompts. The results on

Table 3: **Image classification accuracy for different pretrained objectives** — MAE [90] and MoCo v3 [26] with ViT-Base [23] as backbone. ⋆ denotes the rerun results that calibrate the VPT [4]

| Pretrained objectives | MAE [90] | | | | MoCo v3 [26] | | | |
|---|---|---|---|---|---|---|---|---|
| Methods | Tuned/Total | VTAB-1k [78] [19] | | | Tuned/Total | VTAB-1k [78] [19] | | |
| | | *Natural* [7] | *Specialized* [4] | *Structured* [8] | | Natural [7] | Specialized [4] | Structured [8] |
| Full [CVPR22][92] | 100.00% | 59.31% | 79.68% | 53.82% | 100.00% | 71.95% | 84.72% | 51.98% |
| Linear [CVPR23][92] | 0.04% | 18.87% [0] | 53.72% [0] | 23.70% [0] | 0.04% | 67.46% [4] | 81.08% [0] | 30.33% [0] |
| Partial-1 [NeurIPS14][93] | 8.30% | **58.44%** [5] | **78.28%** [1] | 47.64% [1] | 8.30% | 72.31% [5] | 84.58% [2] | 47.89% [1] |
| Bias [NeurIPS17][30] | 0.16% | 54.55% [1] | 75.68% [1] | **47.70%** [0] | 0.16% | 72.89% [3] | 81.14% [0] | 53.43% [4] |
| Adapter [NeurIPS20][32] | 0.87% | 54.90% [3] | 75.19% [1] | 38.98% [0] | 1.12% | 74.19% [4] | 82.66% [1] | 47.69% [2] |
| VPT-S [ECCV22][4] | 0.05% | 39.96% [1] | 69.65% [0] | 27.50% [0] | 0.06% | 67.34% [3] | 82.26% [0] | 37.55% [0] |
| VPT-D [ECCV22][4] | ⋆ 0.31% | 36.02% [0] | 60.61% [1] | 26.57% [0] | ⋆ 0.22% | 70.27% [4] | 83.04% [0] | 42.38% [0] |
| GPT [ICML23][101] | 0.05% | 47.61% [2] | 76.86% [1] | 36.80% [1] | 0.06% | 74.84% [4] | 83.38% [1] | 49.10% [3] |
| ► Ours | 0.38% | 53.59% [6] {6} | 77.75% [1] {3} | 36.15% [1] {6} | 0.22% | **77.47%** [5] {7} | **85.76%** [3] {4} | **58.74%** [6] {8} |

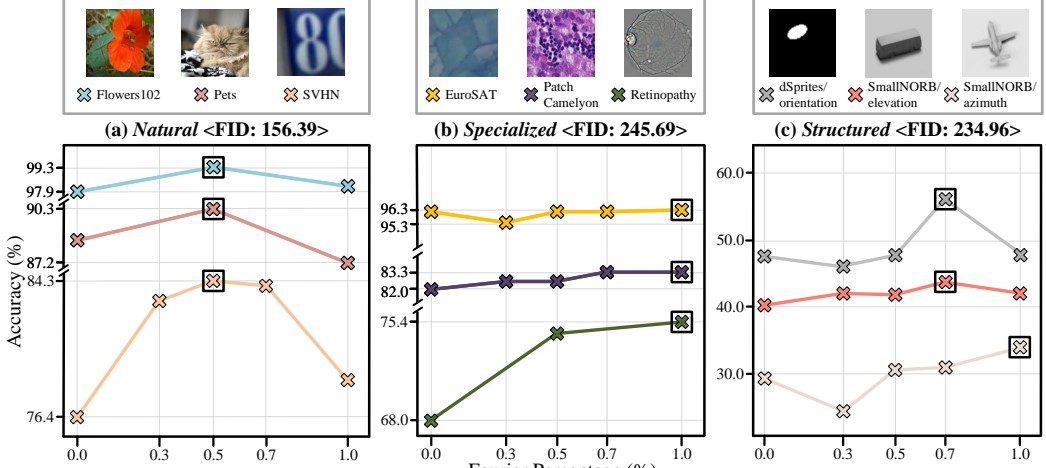

Figure 2: **Image classification accuracy of various Fourier percentages of VTAB-1k [78] for ViT-Base/16 [23].** For better illustration, we randomly select 3 datasets in each group of VTAB-1k. The "Average FID Score of Each Group" is reported in <·>. Our conclusion aligns with **16 of 19** cases. The cross framed by the square indicates the best percentage for each downstream task. Those datasets with only three Fourier percentage reports are due to the prompt length limits.

the ImageNet-21k supervised pretrained Swin-Base [24] are reported in Table 2. It can be seen that VFPT consistently outperforms **all** the other parameter-efficient methods on three VTAB-1k groups. **VFPT on Different Pretraining Objectives.** In Table 3, we report the experimental results on two self-supervised objectives: MAE [90] and MoCo v3 [26]. While VPT yields inconclusive results, VFPT has the **highest** "Number of Wins" compared to full fine-tuning among PEFT methods (*i.e.*, **8 of 19** instances under MAE, and **14 of 19** instances under MoCo v3, respectively). Our method also outperforms VPT by a large margin (*e.g.*, **53.59%** *vs*. 36.02% under MAE on VTAB-1k *Natural*).

♡ *Fourier Contribution.* We conducted experiments to understand the impact of Fourier components by varying the percentages of Fourier prompts in VFPT. As shown in Fig. 2, we observed distinct preferences across the VTAB-1k benchmark, which comprises three groups with varying data disparities (see §4.1). Specifically, the *Natural* group, which has a data distribution similar to the pretrained task (low disparity), shows peak performance when half of the visual prompts are transformed into Fourier prompts, as indicated by the accuracy curves in Fig. 2(a). This suggests that transfer learning is less challenging in this group. Conversely, for the *Specialized* and *Structured* groups, which have data distributions significantly different from the pretrained task (high disparity), the accuracy curves in Fig. 2(b-c) demonstrate that higher classification performance is achieved with an increased percentage of Fourier components. These observations are consistent with our expectations, demonstrating the effectiveness of Fourier prompts in VFPT, especially for tasks with large data disparities. In other words, our approach can be viewed as a generalization of VPT, where the Fourier components learn effective representations from the frequency domain that complement the knowledge from the spatial domain.

## 4.3 Study of Optimization

In this section, we investigate why VFPT achieves better performance and generalization across various tasks from an optimization perspective. Previous works [102] demonstrate that landscape geometry significantly impacts model generalization, so we visualize the loss landscape to

understand the enhanced generality of VFPT. Specifically, in Fig. 3(a), we randomly select two parameter directions for the study, as randomness in directions does not significantly affect the results [102]. There are two key observations supporting the enhanced generality of VFPT. **i) Flatness**: VFPT provides a larger connected region around the local minimum [103] (e.g., ⋆ in the yellow square, where the larger blue area in VFPT offers more optimization choices) and a smoother edge of the loss landscape for mitigating chaotic landscapes (e.g., ● in the green square, where the bumpy contour in VPT is sensitive to loss variations, resulting in

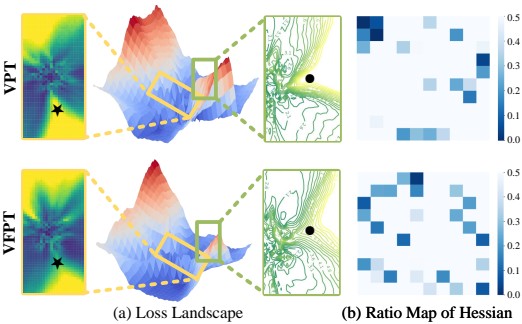

(a) Loss Landscape      (b) Ratio Map of Hessian

Figure 3: Visualization of loss landscape [102] and the ratio map of Hessian [102].

worse generality). This indicates that VFPT achieves a flatter minimizer, which consistently correlates with lower test error [102]. **ii) Convexity**: As eigenvalues of the Hessian directly assess the convexity of a loss function [102], we compute both the maximum and minimum eigenvalues of the Hessian and map their ratios [102]. As shown in Fig. 3(b), a higher prevalence of near-zero negative eigenvalues (in deep blue) in VFPT suggests the presence of more convex regions (25.0% vs. 20.0%) for model optimization. This finding indicates that the incorporation of the Fourier transform in visual prompt tuning effectively mitigates the sharpness of the loss landscape.

## 4.4 Study of Interpretability

To the best of our knowledge, research on the understanding of prompt tuning remains rare [9, 5]. Consequently, our research seeks to both quantitatively and qualitatively examine the impact of Fourier components on the enhancement of visual prompt tuning. For fairness, instead of using enhanced visualization methods [105, 106, 104] that may alter the original expression of the learnable prompts, we visualise and examine the raw average attention head on the last layer of VPT and VFPT.

**Significant attention distribution in learnable prompts**. Observations from both VPT and VFPT in Fig. 4(a) reveal a common phenomenon: there exists a pronounced accumulation of attention scores at learnable prompt locations (*i.e.*, narrow color area on the left side of 2D attention map), indicating that these prompts have a substantial impact on the frozen embeddings during the finetuning stage.

**Global attention scores pattern in Fourier prompts**. We further observe a notably higher concentration in global attention scores when integrating visual Fourier prompts. Specifically, the global attention scores indicate that VFPT also establishes robust correlations within the Transformer's input space [4] (see Fig. 4(a)). In contrast, VPT lacks this correlation, suggesting that it does not adequately consider or integrate

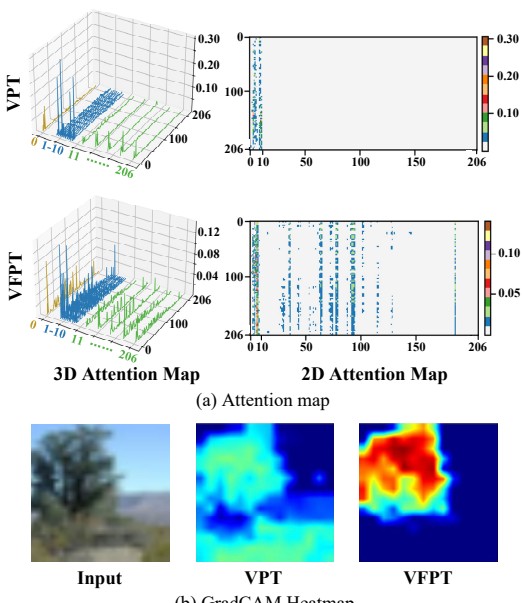

(a) Attention map

Input          VPT          VFPT

(b) GradCAM Heatmap

Figure 4: **Study of interpretability.** (a) The 3D and 2D attention map in VPT and VFPT on a randomly selected sample. The colors ▮, ▮ and ▮ indicate class, prompt and patch tokens, respectively. (b) Corresponding GradCAM [104] maps. Note that red regions correspond to a high score for the class. We present more visualization results in §S4

extensive information from the frozen backbone. Moreover, we find a positive relationship between strong associations and performance gains quantitatively (see §4.2) and qualitatively (see Fig. 4(b)) in VFPT, suggesting that the integration of visual Fourier prompts encourage clear foreground (*i.e.*, tree with high frequency component) - background (*i.e.*, sky with low frequency component) separation.

Table 5: A set of **ablative studies** on VTAB-1k [78] *Natural* and *Specialized* benchmarks in three runs. "Prompt Location" is the placement of the visual Fourier prompts relative to original visual prompts. "Prompt Depth" indicates the layer we use visual Fourier prompts. "Transform Type" is the method we use to transform prompts and input images. "Fourier/Transform Dimension" indicates the dimension we apply using specific transform method. Per-task results are available in Appendix. Same for Table 4.

| Fourier Dimension | | VTAB-1k [78] [19] | |
|---|---|---|---|
| Sequence | Hidden | *Natural* [7] | *Specialized* [4] |
| ✓ | | 80.88% | 83.57% |
| | ✓ | 80.74% | 83.87% |
| ✓ | ✓ | **81.35%** | **84.93%** |

(a) Fourier Prompt Dimension

| Prompt Location | VTAB-1k [78] [19] | |
|---|---|---|
| | *Natural* [7] | *Specialized* [4] |
| $\mathcal{A}$ | 81.02% | 83.80% |
| $\mathcal{R}$ | 78.62% | 82.47% |
| $\mathcal{P}$ | **81.35%** | **84.93%** |

(b) Fourier Prompt Location

| Prompt Depth | VTAB-1k [78] [19] | |
|---|---|---|
| | *Natural* [7] | *Specialized* [4] |
| 1 3 5 7 9 11 | 80.48% | 83.73% |
| 1-6 | 80.79% | 84.34% |
| 7-12 | 80.83% | 83.93% |
| 1-12 | **81.35%** | **84.93%** |

(c) Fourier Prompt Depth

In summary, our findings provide significant insights into the interpretability of prompt tuning, revealing that for both VPT and VFPT, a considerable portion of attention is directed towards the learnable prompts. Further, VFPT exhibit enhanced global feature learning capabilities compared to VPT by interfacing effectively with frozen embeddings, thereby enabling precise capture of distinctive features across diverse downstream tasks. This observation corroborates our findings in §4.2.

## 4.5 Ablation Study

We ablate VFPT's key components on VTAB-1k [78] *Natural* and *Specialized*. More studies are provided in §S2.5.
**Transform Type.** We ablate on other transform method instead to certify the impact of Fourier transform in Table 4, where the Fixed Linear Layer (*i.e.*, FLL) and the Learnable Linear Layer (*i.e.*, LLL) are con-

Table 4: **Ablative studies of transform type** on VTAB-1k [78] *Natural* and *Specialized* benchmarks in three runs. Per-task results are available in Appendix.

| Transform Type (Domain) | Transform Dimension | | VTAB-1k [78] [19] | |
|---|---|---|---|---|
| | Sequence | Hidden | *Natural* [7] | *Specialized* [4] |
| FLL ($\mathcal{S}$) | | ✓ | 80.98% | 84.02% |
| LLL ($\mathcal{S}$) | | ✓ | 80.54% | 82.64% |
| FFT ($\mathcal{F}$) + FDA ($\mathcal{F}$) [71] | ✓ | ✓ | 80.90% | 84.03% |
| FFT ($\mathcal{F}$) | ✓ | ✓ | **81.35%** | **84.93%** |

sidered. Compared with FFT, a fixed non-parameter Fourier domain transform in sequence and hidden dimension, the FLL operation considers only a fixed spatial domain transform in hidden dimension; the LLL further unfixes the transformation to enable gradient updates. As seen, both FLL and LLL show inferior performance to FFT. We further consider the impact of current Fourier domain adaption approach [71], which maps a source image to a target "style" without altering semantic content. However, no significant improvement can be observed.
**Fourier Prompt Dimension.** A fundamental distinction between VFPT and other methods is the incorporation of FFT into visual prompts. In our standard implementation, we utilize 2D FFTs across both sequence length and hidden dimensions. Here, we explore the impact of each dimension's transformation individually. As shown in Table 5(a), the separate Fourier transformations along each dimension appear to have similar contributions (*i.e.*, 80.88% *vs*. 80.74% in *Natural*). However, the combined application of transformations across both dimensions (*i.e.*, 2D FFTs) demonstrates a synergistic effect, yielding significant improvement in performance.
**Fourier Prompt Location.** In Table 5(b), three prompt locations are considered for VFPT, which are "Prepend" (*i.e.*, $\mathcal{P}$), "Append" (*i.e.*, $\mathcal{A}$), and "Random" (*i.e.*, $\mathcal{R}$). Specifically, $\mathcal{P}$ and $\mathcal{A}$ prepend visual Fourier prompts before or after visual prompts, and $\mathcal{R}$ randomly selects the position for visual Fourier prompts in each layer. As seen, both $\mathcal{P}$ and $\mathcal{A}$ show competitive results, validating the robustness of VFPT *w.r.t.* prompt locations. In alignment with the findings in [5, 4], we choose $\mathcal{P}$ as our baseline method in all experiments since it reaches superior results (*i.e.*, 81.35% *vs* 81.02% in *Natural*).
**Fourier Prompt Depth.** Table 5(c) presents the performance of VFPT based on the specific layer at which visual Fourier prompts are employed. The results suggest that employment on separate layers also yields a accuracy improvement compared with VPT. Further application of visual Fourier prompts across all layers fosters the best overall performance.

## 5 Conclusion

We present **V**isual **F**ourier **P**rompt **T**uning (**VFPT**), a simple yet powerful parameter-efficient visual prompt tuning approach that draws insights from human visual cognition. It has merits in: **i)** integrating spatial and frequency domain information through an intuitive yet effective design; **ii)** demonstrating generality across datasets with varying disparities while ensuring powerful performance; and **iii)** thoroughly investigating the associations between learnable prompts and frozen embeddings to elucidate this generality. As a whole, we conclude that the outcomes elucidated in this paper impart essential understandings and necessitate further exploration within this realm.

## 6 Acknowledgements

This research was supported by the National Science Foundation under Grant No. 2242243.

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

- §S1 provides **per-task results on VTAB-1k and FGVC** image classification benchmarks with confidence analysis, where the overall results have been provided in the main paper.
- §S2 provides **per-task results on ablation study**, where the overall results have been provided in the main paper. Further study of sensitivity of Fourier prompt percentages and prompt lengths is included in §S2.5.
- §S3 provides **per-task results on Fourier percentage**, where partial results have been provided in the main paper.
- §S4 presents more details and results of **visualization of attention maps**.
- §S4 presents more details and results of **visualization of loss landscapes**.
- §S6 discusses our potential **extension to language tasks**.
- §S7 further analyze the **complexity** of our approach.
- §S8 shows related **asset license and consent** to our work.
- §S9 claims **reproducibility** of our approach.
- §S10 discusses the **social impact** of our research.
- §S11 adds more **discussions**, and points out potential directions of our **future work**.

# S1   Per-task Results on VTAB-1k and FGVC

## S1.1   Per-task Results on ViT-Base

To provide comprehensive results from the paper, we report the average per-task test accuracy (*i.e.*, 3 runs, 24 tasks) on VTAB-1k [78] *Natural*, *Specialized* and *Structured*, respectively (see Table S1, S2 and S3). We also report per-task FGVC [4] results (5 tasks) in Table S4. VPT-SHALLOW [4] is also included for completeness (*i.e.*, VPT-SHALLOW only introduces 1-st layer visual prompts). In conclusion, VFPT shows consistently better performance in various downstream tasks.

Table S1: **VTAB-1k [78] *Natural* per-task results for ViT-Base/16 [23]** pretrained on supervised ImageNet-21k. Consistent to our paper, "Number of Wins" in [·] compared to full fine-tuning [92]. "Tuned/Total" is the percentage of tuned parameters in each task, along with the average results of those percentages in each group. The highest accuracy among all approaches except FULL are shown in **bold**. † denotes method using soft filtered prompts to reduce the parameter usage in learnable visual prompts, requiring specialized devices to facilitate acceleration. All results are averaged in three runs with different initialization seeds. Same for Table S2-S21. We also report standard deviation error bars for our main results (Table S1, S2, S3 and S4) by calculating each task respectively and averaging across them. Other tables show similar trends on standard deviation error bars.

| ViT-Base/16 [23] (85.8M) | VTAB-1k [78] *Natural* [7] | | | | | | | Mean |
|---|---|---|---|---|---|---|---|---|
| | CIFAR-100 | Caltech101 | DTD | Flowers102 | Pets | SVHN | Sun397 | |
| FULL [92] | 68.9 | 87.7 | 64.3 | 97.2 | 86.9 | 87.4 | 38.8 | 75.88 |
| LINEAR [92] | 63.4 | 85.0 | 63.2 | 97.0 | 86.3 | 36.6 | 51.0 | 68.93 [1] |
| PARTIAL-1 [93] | 66.8 | 85.9 | 62.5 | 97.3 | 85.5 | 37.6 | 50.6 | 69.44 [2] |
| MLP-2 [94] | 63.2 | 84.8 | 60.5 | 97.6 | 85.9 | 34.1 | 47.8 | 67.70 [2] |
| MLP-3 [94] | 63.8 | 84.7 | 62.3 | 97.4 | 84.7 | 32.5 | 49.2 | 67.80 [2] |
| MLP-5 [94] | 59.3 | 84.4 | 59.9 | 96.1 | 84.4 | 30.9 | 46.8 | 65.98 [1] |
| MLP-9 [94] | 53.1 | 80.5 | 53.9 | 95.1 | 82.6 | 24.4 | 43.7 | 61.90 [1] |
| SIDETUNE [31] | 60.7 | 60.8 | 53.6 | 95.5 | 66.7 | 34.9 | 35.3 | 58.21 [0] |
| BIAS [30] | 72.8 | 87.0 | 59.2 | 97.5 | 85.3 | 59.9 | 51.4 | 73.30 [3] |
| ADAPTER-256 [32] | 74.1 | 86.1 | 63.2 | 97.7 | 87.0 | 34.6 | 50.8 | 70.50 [4] |
| ADAPTER-64 [32] | 74.2 | 85.8 | 62.7 | 97.6 | 87.2 | 36.3 | 50.9 | 70.65 [4] |
| ADAPTER-8 [32] | 74.2 | 85.7 | 62.7 | 97.8 | 87.2 | 36.4 | 50.7 | 70.67 [4] |
| VPT-SHALLOW [4] | 77.7 | 86.9 | 62.6 | 97.5 | 87.3 | 74.5 | 51.2 | 76.81 [4] |
| - Tuned / Total (%) | 0.18 | 0.10 | 0.04 | 0.27 | 0.08 | 0.19 | 0.36 | 0.17 |
| VPT-DEEP [4] | 78.8 | 90.8 | 65.8 | 98.0 | 88.3 | 78.1 | 49.6 | 78.48 [6] |
| - Tuned / Total (%) | 0.20 | 0.20 | 0.15 | 0.10 | 0.04 | 0.54 | 0.41 | 0.23 |
| † E2VPT [5] | 78.6 | 89.4 | 67.8 | 98.2 | 88.5 | 85.3 | 52.3 | 80.01 [6] |
| - Tuned / Total (%) | 0.22 | 0.19 | 0.12 | 0.11 | 0.05 | 0.24 | 0.43 | 0.19 |
| OURS | **80.7** ± (0.15) | **91.4** ± (0.11) | **69.4** ± (0.27) | **99.3** ± (0.05) | **90.3** ± (0.29) | **85.6** ± (0.95) | **52.7** ± (0.47) | **81.35** ± (0.33) [6] |
| - Tuned / Total (%) | 0.20 | 0.31 | 0.20 | 0.11 | 0.06 | 0.12 | 0.41 | 0.21 |
| - Fourier Percentage (%) | 70.0 | 50.0 | 30.0 | 50.0 | 50.0 | 20.0 | 50.0 | 45.7 |

Table S2: **VTAB-1k [78]** *Specialized* **per-task results for ViT-Base/16 [23]** pretrained on supervised ImageNet-21k.

| ViT-Base/16 [23] (85.8M) | VTAB-1k [78] *Specialized* (4) | | | | Mean |
|---|---|---|---|---|---|
| | Patch Camelyon | EuroSAT | Resisc45 | Retinopathy | |
| FULL [92] | 79.7 | 95.7 | 84.2 | 73.9 | 83.36 |
| LINEAR [92] | 78.5 | 87.5 | 68.6 | 74.0 | 77.16 [1] |
| PARTIAL-1 [93] | 78.6 | 89.8 | 72.5 | 73.3 | 78.53 [0] |
| MLP-2 [94] | 74.3 | 88.8 | 67.1 | 73.2 | 75.86 [0] |
| MLP-3 [94] | 77.0 | 88.0 | 70.2 | 56.1 | 72.83 [0] |
| MLP-5 [94] | 73.7 | 87.2 | 64.8 | 71.5 | 74.31 [0] |
| MLP-9 [94] | 78.5 | 83.0 | 60.2 | 72.3 | 73.49 [0] |
| SIDETUNE [31] | 58.5 | 87.7 | 65.2 | 61.0 | 68.12 [0] |
| BIAS [30] | 78.7 | 91.6 | 72.9 | 69.8 | 78.25 [0] |
| ADAPTER-256 [32] | 76.3 | 88.0 | 73.1 | 70.5 | 76.98 [0] |
| ADAPTER-64 [32] | 76.3 | 87.5 | 73.7 | 70.9 | 77.10 [0] |
| ADAPTER-8 [32] | 76.9 | 89.2 | 73.5 | 71.6 | 77.80 [0] |
| VPT-SHALLOW [4] | 78.2 | 92.0 | 75.6 | 72.9 | 79.66 [0] |
| - Tuned / Total (%) | 0.01 | 0.05 | 0.09 | 0.01 | 0.04 |
| VPT-DEEP [4] | 81.8 | 96.1 | 83.4 | 68.4 | 82.43 [2] |
| - Tuned / Total (%) | 1.06 | 1.07 | 0.15 | 0.02 | 0.57 |
| † E2VPT [5] | 82.5 | **96.8** | **84.8** | 73.6 | 84.43 [3] |
| - Tuned / Total (%) | 0.20 | 0.29 | 0.12 | 0.07 | 0.17 |
| OURS | **83.5** ± (0.09) | 96.5 ± (0.06) | 84.4 ± (0.36) | **75.4** ± (0.05) | **84.93** ± (0.14)[4] |
| - Tuned / Total (%) | 1.06 | 0.12 | 0.11 | 0.03 | 0.33 |
| - Fourier Percentage (%) | 100.0 | 30.0 | 100.0 | 100.0 | 82.5 |

Table S3: **VTAB-1k [78]** *Structured* **per-task results for ViT-Base/16 [23]** pretrained on supervised ImageNet-21k.

| ViT-Base/16 [23] (85.8M) | VTAB-1k [78] *Structured* [8] | | | | | | | | Mean |
|---|---|---|---|---|---|---|---|---|---|
| | Clevr/ count | Clevr/ distance | DMLab | KITTI/ distance | dSprites/ location | dSprites/ orientation | SmallNORB/ azimuth | SmallNORB/ elevation | |
| FULL [92] | 56.3 | 58.6 | 41.7 | 65.5 | 57.5 | 46.7 | 25.7 | 29.1 | 47.64 |
| LINEAR [92] | 34.3 | 30.6 | 33.2 | 55.4 | 12.5 | 20.0 | 9.6 | 19.2 | 26.84 [0] |
| PARTIAL-1 [93] | 41.5 | 34.3 | 33.9 | 61.0 | 31.3 | 32.8 | 16.3 | 22.4 | 34.17 [0] |
| MLP-2 [94] | 45.2 | 31.6 | 31.8 | 55.7 | 30.9 | 24.6 | 16.6 | 23.3 | 32.47 [0] |
| MLP-3 [94] | 47.8 | 32.8 | 32.3 | 58.1 | 12.9 | 21.2 | 15.2 | 24.8 | 30.62 [0] |
| MLP-5 [94] | 50.8 | 32.3 | 31.5 | 56.4 | 7.5 | 20.8 | 14.4 | 20.4 | 29.23 [0] |
| MLP-9 [94] | 47.5 | 27.9 | 28.9 | 54.0 | 6.2 | 17.7 | 10.8 | 16.2 | 26.15 [0] |
| SIDETUNE [31] | 27.6 | 22.6 | 31.3 | 51.7 | 8.2 | 14.4 | 9.8 | 21.8 | 23.41 [0] |
| BIAS [30] | 61.5 | 55.6 | 32.4 | 55.9 | 66.6 | 40.0 | 15.7 | 25.1 | 44.09 [2] |
| ADAPTER-256 [32] | 45.7 | 37.4 | 31.2 | 53.2 | 30.3 | 25.4 | 13.8 | 22.1 | 32.39 [0] |
| ADAPTER-64 [32] | 42.9 | 39.9 | 30.4 | 54.5 | 31.9 | 25.6 | 13.5 | 21.4 | 32.51 [0] |
| ADAPTER-8 [32] | 45.2 | 41.8 | 31.1 | 56.4 | 30.4 | 24.6 | 13.2 | 22.0 | 33.09 [0] |
| VPT-SHALLOW [4] | 50.5 | 58.6 | 40.5 | 67.1 | 68.7 | 36.1 | 20.2 | 34.1 | 46.98 [4] |
| - Tuned / Total (%) | 0.10 | 0.18 | 0.09 | 0.09 | 0.10 | 0.10 | 0.19 | 0.19 | 0.13 |
| VPT-DEEP [4] | 68.5 | 60.0 | 46.5 | 72.8 | 73.6 | 47.9 | 32.9 | 37.8 | 54.98 [8] |
| - Tuned / Total (%) | 0.54 | 2.11 | 1.07 | 0.54 | 0.12 | 0.55 | 2.12 | 2.11 | 1.14 |
| † E2VPT [5] | 71.7 | 61.2 | 47.9 | 75.8 | 80.8 | 48.1 | 31.7 | 41.9 | 57.39 [8] |
| - Tuned / Total (%) | 0.34 | 0.65 | 0.44 | 0.36 | 0.12 | 0.38 | 1.14 | 0.66 | 0.51 |
| OURS | **75.8** ± (0.94) | **63.2** ± (0.51) | **48.3** ± (0.93) | **79.3** ± (0.38) | **81.5** ± (1.06) | **56.0** ± (0.51) | **34.1** ± (1.05) | **43.4** ± (0.42) | **60.19** ± (0.72)[8] |
| - Tuned / Total (%) | 0.54 | 2.11 | 0.11 | 0.71 | 0.12 | 0.55 | 1.91 | 2.11 | 1.02 |
| - Fourier Percentage (%) | 100.0 | 100.0 | 70.0 | 50.0 | 100.0 | 70.0 | 100.0 | 70.0 | 82.5 |

Table S4: **FGVC [4] per-task results for ViT-Base/16 [23]** pretrained on supervised ImageNet-21k.

| ViT-Base/16 [23] (85.8M) | FGVC [4] [5] | | | | | Mean |
|---|---|---|---|---|---|---|
| | CUB-200-2011 | NAbirds | Oxford Flowers | Stanford Dogs | Stanford Cars | |
| FULL [92] | 87.3 | 82.7 | 98.8 | 89.4 | 84.5 | 88.54 |
| LINEAR [92] | 85.3 | 75.9 | 97.9 | 86.2 | 51.3 | 79.32 [0] |
| PARTIAL-1 [93] | 85.6 | 77.8 | 98.2 | 85.5 | 66.2 | 82.63 [0] |
| MLP-2 [94] | 85.7 | 77.2 | 98.2 | 85.4 | 54.9 | 80.28 [0] |
| MLP-3 [94] | 85.1 | 77.3 | 97.9 | 84.9 | 53.8 | 79.80 [0] |
| MLP-5 [94] | 84.2 | 76.7 | 97.6 | 84.8 | 50.2 | 78.71 [0] |
| MLP-9 [94] | 83.2 | 76.0 | 96.2 | 83.7 | 47.6 | 77.31 [0] |
| SIDETUNE [31] | 84.7 | 75.8 | 96.9 | 85.8 | 48.6 | 78.35 [0] |
| BIAS [30] | 88.4 | 84.2 | 98.8 | 91.2 | 79.4 | 88.41 [3] |
| ADAPTER-256 [32] | 87.2 | 84.3 | 98.5 | 89.9 | 68.6 | 85.70 [2] |
| ADAPTER-64 [32] | 87.1 | 84.3 | 98.5 | 89.8 | 68.6 | 85.67 [2] |
| ADAPTER-8 [32] | 87.3 | 84.3 | 98.4 | 88.8 | 68.4 | 85.46 [1] |
| VPT-SHALLOW [4] | 86.7 | 78.8 | 98.4 | 90.7 | 68.7 | 84.62 [1] |
| - Tuned / Total (%) | 0.31 | 0.54 | 0.23 | 0.20 | 0.26 | 0.31 |
| VPT-DEEP [4] | 88.5 | 84.2 | 99.0 | 90.2 | **83.6** | 89.11 [4] |
| - Tuned / Total (%) | 0.29 | 1.02 | 0.14 | 1.17 | 2.27 | 0.98 |
| † E2VPT [5] | **89.1** | **84.6** | 99.1 | **90.5** | 82.8 | 89.22 [4] |
| - Tuned / Total (%) | 0.32 | 0.65 | 0.15 | 0.88 | 1.27 | 0.65 |
| Ours | 88.7 ± (0.02) | 84.5 ± (0.01) | **99.1** ± (0.01) | 90.4 ± (0.13) | 83.6 ± (0.04) | **89.24** ± (0.04) [4] |
| - Tuned / Total (%) | 0.29 | 1.02 | 0.15 | 1.17 | 2.27 | 0.98 |
| - Fourier Percentage (%) | 50.0 | 50.0 | 30.0 | 50.0 | 0.0 | 36.0 |

## S1.2 Per-task Results on Swin-Base

Table S5: **VTAB-1k [78]** *Natural* **per-task results for Swin-Base [24]** pretrained on supervised ImageNet-21k. Specially, the highest accuracy is shown in **bold**. Same for Table S6 and S7

| Swin-Base [24] (86.7M) | VTAB-1k [78] *Natural* (7) | | | | | | | Mean |
|---|---|---|---|---|---|---|---|---|
| | CIFAR-100 | Caltech101 | DTD | Flowers102 | Pets | SVHN | Sun397 | |
| FULL [92] | 72.2 | 88.0 | 71.2 | 98.3 | 89.5 | 89.4 | 45.0 | 79.10 |
| VPT-SHALLOW [4] | 77.7 | 86.9 | 62.6 | 97.5 | 87.3 | 74.5 | 51.2 | 76.81 [4] |
| - Tuned / Total (%) | 0.18 | 0.10 | 0.04 | 0.27 | 0.08 | 0.19 | 0.36 | 0.17 |
| VPT-DEEP [4] | 79.6 | 90.8 | 78.0 | 99.5 | **91.4** | 46.4 | 51.7 | 78.78 [6] |
| - Tuned / Total (%) | 0.13 | 0.13 | 0.07 | 0.70 | 0.06 | 0.70 | 0.48 | 0.28 |
| † E2VPT [5] | 82.9 | 92.4 | **78.5** | 99.6 | 91.4 | 82.2 | 56.2 | 83.31 [6] |
| - Tuned / Total (%) | 0.27 | 0.15 | 0.08 | 0.15 | 0.07 | 0.44 | 0.49 | 0.24 |
| OURS | **83.9** | **93.0** | 77.9 | 99.6 | 91.4 | **89.5** | **56.4** | **84.53** [7] |
| - Tuned / Total (%) | 0.15 | 0.15 | 0.13 | 0.15 | 0.07 | 0.70 | 0.49 | 0.26 |
| - Fourier Percentage (%) | 100.0 | 100.0 | 100.0 | 100.0 | 100.0 | 100.0 | 100.0 | 100.0 |

Table S6: **VTAB-1k [78]** *Specialized* **per-task results for Swin-Base [24]** pretrained on supervised ImageNet-21k.

| Swin-Base [24] (86.7M) | VTAB-1k [78] *Specialized* [4] | | | | Mean |
|---|---|---|---|---|---|
| | Patch Camelyon | EuroSAT | Resisc45 | Retinopathy | |
| FULL [92] | **86.6** | 96.9 | **87.7** | 73.6 | 86.21 |
| VPT-SHALLOW [4] | 78.2 | 92.0 | 75.6 | 72.9 | 79.66 [0] |
| - Tuned / Total (%) | 0.01 | 0.05 | 0.09 | 0.01 | 0.04 |
| VPT-DEEP [4] | 80.1 | 96.2 | 85.0 | 72.0 | 83.33 [0] |
| - Tuned / Total (%) | 0.07 | 0.13 | 0.19 | 0.02 | 0.10 |
| † E2VPT [5] | 83.8 | 97.2 | 84.8 | 74.0 | 84.95 [2] |
| - Tuned / Total (%) | 0.09 | 0.04 | 0.20 | 0.03 | 0.09 |
| OURS | 86.3 | **97.3** | 86.9 | **74.1** | **86.15** [2] |
| - Tuned / Total (%) | 0.07 | 0.15 | 0.19 | 0.03 | 0.11 |
| - Fourier Percentage (%) | 100.0 | 100.0 | 50.0 | 100.0 | 87.5 |

Table S7: **VTAB-1k [78]** *Structured* **per-task results for Swin-Base [24]** pretrained on supervised ImageNet-21k.

| Swin-Base [24] (86.7M) | VTAB-1k [78] *Structured* [8] | | | | | | | | Mean |
|---|---|---|---|---|---|---|---|---|---|
| | Clevr/ count | Clevr/ distance | DMLab | KITTI/ distance | dSprites/ location | dSprites/ orientation | SmallNORB/ azimuth | SmallNORB/ elevation | |
| FULL [92] | **75.7** | 59.8 | **54.6** | 78.6 | 79.4 | **53.6** | **34.6** | **40.9** | 59.65 |
| VPT-SHALLOW [4] | 50.5 | 58.6 | 40.5 | 67.1 | 68.7 | 36.1 | 20.2 | 34.1 | 46.98 [4] |
| - Tuned / Total (%) | 0.10 | 0.18 | 0.09 | 0.09 | 0.10 | 0.10 | 0.19 | 0.19 | 0.13 |
| VPT-DEEP [4] | 67.6 | 59.4 | 50.1 | 61.3 | 74.4 | 50.6 | 25.7 | 25.7 | 51.85 [0] |
| - Tuned / Total (%) | 0.70 | 0.70 | 0.14 | 0.69 | 0.15 | 0.09 | 0.16 | 0.02 | 0.38 |
| † E2VPT [5] | 74.0 | 61.2 | 49.5 | **81.0** | 80.3 | 50.7 | 27.9 | 34.2 | 57.35 [3] |
| - Tuned / Total (%) | 0.70 | 0.43 | 0.14 | 0.51 | 0.17 | 0.17 | 0.16 | 0.04 | 0.29 |
| OURS | 74.9 | **61.5** | 50.0 | 80.5 | **82.7** | 50.6 | 29.9 | 35.6 | **58.21** [3] |
| - Tuned / Total (%) | 0.70 | 0.70 | 0.15 | 0.92 | 0.16 | 0.09 | 0.16 | 0.04 | 0.36 |
| - Fourier Percentage (%) | 100.0 | 50.0 | 100.0 | 100.0 | 100.0 | 100.0 | 100.0 | 50.0 | 87.5 |

## S1.3 Per-task Results on MAE and MoCo v3

Table S8: **VTAB-1k [78]** *Natural* **per-task results for ViT-Base/16 [23]** pretrained on MAE [90]. Since VPT [4] have considerably lower performance, we do not list the per-task results for simplicity. We instead compare our method to full fine-tuning, and the highest accuracy is shown in **bold**. We post the "Number of Wins" in [·] to full fine-tuning (FULL) [92]. Same for Table S9-S13.

| ViT-Base/16 [23] (85.8M) | VTAB-1k [78] *Natural* [7] | | | | | | | Mean |
|---|---|---|---|---|---|---|---|---|
| | CIFAR-100 | Caltech101 | DTD | Flowers102 | Pets | SVHN | Sun397 | |
| FULL [92] | 24.6 | 84.2 | 56.9 | 72.7 | 74.4 | **86.6** | 15.8 | 59.31 |
| OURS | **36.0** | **87.7** | **58.0** | **74.3** | **76.3** | 19.6 | **23.3** | **53.59** [6] |
| - Tuned / Total (%) | 0.13 | 0.11 | 0.06 | 0.11 | 0.06 | 1.07 | 0.38 | 0.28 |
| - Fourier Percentage (%) | 100.0 | 50.0 | 50.0 | 100.0 | 50.0 | 100.0 | 100.0 | 75.6 |

Table S9: **VTAB-1k [78]** *Specialized* **per-task results for ViT-Base/16 [23]** pretrained on MAE [90].

| ViT-Base/16 [23] (85.8M) | VTAB-1k [78] *Specialized* [4] | | | | Mean |
|---|---|---|---|---|---|
| | Patch Camelyon | EuroSAT | Resisc45 | Retinopathy | |
| FULL [92] | **81.8** | **94.0** | **72.3** | 70.6 | **79.68** |
| OURS | 76.9 | 91.3 | 69.2 | **73.6** | 77.75 [1] |
| - Tuned / Total (%) | 0.06 | 0.03 | 0.13 | 0.54 | 0.17 |
| - Fourier Percentage (%) | 50.0 | 100.0 | 50.0 | 50.0 | 62.5 |

Table S10: **VTAB-1k [78]** *Strcutured* **per-task results for ViT-Base/16 [23]** pretrained on MAE [90].

| ViT-Base/16 [23] (85.8M) | Clevr/ count | Clevr/ distance | DMLab | KITTI/ distance | dSprites/ location | dSprites/ orientation | SmallNORB/ azimuth | SmallNORB/ elevation | Mean |
|---|---|---|---|---|---|---|---|---|---|
| FULL [92] | **67.0** | **59.8** | **45.2** | 75.3 | **72.5** | **47.5** | **30.2** | **33.0** | **53.82** |
| OURS | 47.6 | 45.3 | 40.7 | **80.7** | 13.7 | 34.6 | 9.3 | 17.3 | 36.15 [1] |
| - Tuned / Total (%) | 0.03 | 2.11 | 0.03 | 0.20 | 2.12 | 0.04 | 0.04 | 0.12 | 0.58 |
| - Fourier Percentage (%) | 50.0 | 100.0 | 100.0 | 50.0 | 50.0 | 50.0 | 100.0 | 50.0 | 68.8 |

Table S11: **VTAB-1k [78]** *Natural* **per-task results for ViT-Base/16 [23]** pretrained on MOCO [26].

| ViT-Base/16 [23] (85.8M) | VTAB-1k [78] *Natural* [7] | | | | | | | Mean |
|---|---|---|---|---|---|---|---|---|
| | CIFAR-100 | Caltech101 | DTD | Flowers102 | Pets | SVHN | Sun397 | |
| FULL [92] | 57.6 | **91.0** | 64.6 | 91.6 | 79.9 | **89.8** | 29.1 | 71.95 |
| OURS | **73.6** | 90.5 | **70.5** | **92.4** | **88.3** | 84.7 | **42.3** | **77.47** [5] |
| - Tuned / Total (%) | 0.20 | 1.15 | 0.06 | 0.11 | 0.14 | 0.06 | 0.46 | 0.31 |
| - Fourier Percentage (%) | 50.0 | 100.0 | 50.0 | 50.0 | 100.0 | 100.0 | 50.0 | 71.4 |

Table S12: **VTAB-1k [78]** *Specialized* **per-task results for ViT-Base/16 [23]** pretrained on MOCO [26].

| ViT-Base/16 [23] (85.8M) | VTAB-1k [78] *Specialized* [4] | | | | Mean |
|---|---|---|---|---|---|
| | Patch Camelyon | EuroSAT | Resisc45 | Retinopathy | |
| FULL [92] | 85.1 | **96.4** | 83.1 | 74.2 | 84.72 |
| OURS | **86.7** | 95.7 | **85.2** | **75.5** | **85.76** [3] |
| - Tuned / Total (%) | 0.11 | 0.03 | 0.15 | 0.06 | 0.09 |
| - Fourier Percentage (%) | 100.0 | 100.0 | 50.0 | 50.0 | 75.0 |

Table S13: **VTAB-1k [78]** *Structured* **per-task results for ViT-Base/16 [23]** pretrained on MOCO [26].

| ViT-Base/16 [23] (85.8M) | Clevr/ count | Clevr/ distance | DMLab | KITTI/ distance | dSprites/ location | dSprites/ orientation | SmallNORB/ azimuth | SmallNORB/ elevation | Mean |
|---|---|---|---|---|---|---|---|---|---|
| FULL [92] | 55.2 | 56.9 | 44.6 | 77.9 | 63.8 | **49.0** | **31.5** | 36.9 | 51.98 |
| OURS | **76.3** | **63.0** | **46.1** | **82.2** | **85.3** | 47.4 | 23.8 | **45.8** | **58.74** [6] |
| - Tuned / Total (%) | 0.06 | 1.07 | 0.06 | 0.23 | 0.12 | 0.07 | 0.07 | 0.06 | 0.22 |
| - Fourier Percentage (%) | 50.0 | 50.0 | 50.0 | 50.03 | 50.0 | 50.0 | 100.0 | 50.0 | 56.3 |

# S2    Per-task Results on Ablation Study

## S2.1    Per-task Results of Transform Type on VTAB-1k *Natural* and *Specialized*

Table S14: **Transform type per-task results on VTAB-1k [78]** *Natural* for ViT-Base/16 [23] pretrained on supervised ImageNet-21k.

| ViT-Base/16 [23]
(85.8M) | VTAB-1k [78] *Natural* [7] | | | | | | | Mean |
|---|---|---|---|---|---|---|---|---|
| | CIFAR-100 | Caltech101 | DTD | Flowers102 | Pets | SVHN | Sun397 | |
| FULL [92] | 57.6 | 91.0 | 64.6 | 91.6 | 79.9 | 89.8 | 29.1 | 71.95 |
| VPT-SHALLOW [4] | 77.7 | 86.9 | 62.6 | 97.5 | 87.3 | 74.5 | 51.2 | 76.81 [4] |
| - Tuned / Total (%) | 0.18 | 0.10 | 0.04 | 0.27 | 0.08 | 0.19 | 0.36 | 0.17 |
| VPT-DEEP [4] | 78.8 | 90.8(3) | 65.8 | 98.0 | 88.3 | 78.1 | 49.6 | 78.48 [6] |
| - Tuned / Total (%) | 0.20 | 0.20 | 0.15 | 0.10 | 0.04 | 0.54 | 0.41 | 0.23 |
| OURS-FLL | **80.8** | **91.7** | **70.5** | 98.5 | 89.4 | 83.3 | 52.7 | 80.98 [6] |
| - Tuned / Total (%) | 0.20 | 0.31 | 0.20 | 0.11 | 0.06 | 0.12 | 0.41 | 0.21 |
| OURS-LLL | 79.5 | 91.5 | 70.1 | 98.5 | 89.6 | 82.0 | 52.6 | 80.54 [6] |
| - Tuned / Total (%) | 0.20 | 0.31 | 0.20 | 0.11 | 0.06 | 0.12 | 0.41 | 0.21 |
| OURS-FFT + FDA [71] | 80.7 | 91.4 | 69.4 | 98.5 | 89.9 | 83.6 | 52.7 | 80.90 [6] |
| - Tuned / Total (%) | 0.20 | 0.31 | 0.20 | 0.11 | 0.06 | 0.12 | 0.41 | 0.21 |
| OURS-FFT (default) | 80.7 | 91.4 | 69.4 | **99.3** | **90.3** | **85.6** | 52.7 | **81.35** [6] |
| - Tuned / Total (%) | 0.20 | 0.31 | 0.20 | 0.11 | 0.06 | 0.12 | 0.41 | 0.21 |

Table S15: **Transform type per-task results on VTAB-1k [78]** *Specialized* for ViT-Base/16 [23] pretrained on supervised ImageNet-21k.

| ViT-Base/16 [23]
(85.8M) | VTAB-1k [78] *Specialized* [4] | | | | Mean |
|---|---|---|---|---|---|
| | Patch Camelyon | EuroSAT | Resisc45 | Retinopathy | |
| FULL [92] | 85.1 | 96.4 | 83.1 | 74.3 | 84.72 |
| VPT-SHALLOW [4] | 78.2 | 92.0 | 75.6 | 72.9 | 79.66 [0] |
| - Tuned / Total (%) | 0.01 | 0.05 | 0.09 | 0.01 | 0.04 |
| VPT-DEEP [4] | 81.8 | 96.1 | 83.4 | 68.4 | 82.43 [2] |
| - Tuned / Total (%) | 1.06 | 1.07 | 0.15 | 0.02 | 0.57 |
| OURS-FLL | 83.3 | 95.2 | 83.5 | 74.1 | 84.02 [3] |
| - Tuned / Total (%) | 1.06 | 0.12 | 0.11 | 0.03 | 0.33 |
| OURS-LLL | 77.3 | 95.5 | 82.7 | 75.0 | 82.64 [3] |
| - Tuned / Total (%) | 1.06 | 0.12 | 0.11 | 0.03 | 0.33 |
| OURS-FFT + FDA [71] | 83.2 | 95.1 | 82.4 | 75.4 | 84.03 [3] |
| - Tuned / Total (%) | 1.06 | 0.12 | 0.11 | 0.03 | 0.33 |
| OURS-FFT (default) | **83.5** | **96.5** | **84.4** | 75.4 | **84.93** [4] |
| - Tuned / Total (%) | 1.06 | 0.12 | 0.11 | 0.03 | 0.33 |

## S2.2    Per-task Results of Fourier Prompt Depth on VTAB-1k *Natural* and *Specialized*

Table S16: **Fourier prompt depth per-task results on VTAB-1k [78]** *Natural* for ViT-Base/16 [23] pretrained on supervised ImageNet-21k.

| ViT-Base/16 [23]
(85.8M) | VTAB-1k [78] *Natural* [7] | | | | | | | Mean |
|---|---|---|---|---|---|---|---|---|
| | CIFAR-100 | Caltech101 | DTD | Flowers102 | Pets | SVHN | Sun397 | |
| FULL [92] | 57.6 | 91.0 | 64.6 | 91.6 | 79.9 | 89.8 | 29.1 | 71.95 |
| VPT-SHALLOW [4] | 77.7 | 86.9 | 62.6 | 97.5 | 87.3 | 74.5 | 51.2 | 76.81 [4] |
| - Tuned / Total (%) | 0.18 | 0.10 | 0.04 | 0.27 | 0.08 | 0.19 | 0.36 | 0.17 |
| VPT-DEEP [4] | 78.8 | 90.8(3) | 65.8 | 98.0 | 88.3 | 78.1 | 49.6 | 78.48 [6] |
| - Tuned / Total (%) | 0.20 | 0.20 | 0.15 | 0.10 | 0.04 | 0.54 | 0.41 | 0.23 |
| OURS (1 3 5 7 9 11) | 80.0 | 91.6 | 68.4 | 98.5 | 89.5 | 82.7 | 52.7 | 80.48 [6] |
| - Tuned / Total (%) | 0.20 | 0.31 | 0.20 | 0.11 | 0.06 | 0.12 | 0.41 | 0.21 |
| OURS (1-6) | **80.8** | **91.8** | 69.5 | 98.5 | 89.4 | 83.5 | 52.0 | 80.79 [6] |
| - Tuned / Total (%) | 0.20 | 0.31 | 0.20 | 0.11 | 0.06 | 0.12 | 0.41 | 0.21 |
| OURS (7-12) | 80.3 | 91.1 | **70.0** | 98.6 | 89.4 | 83.6 | 52.7 | 80.83 [6] |
| - Tuned / Total (%) | 0.20 | 0.31 | 0.20 | 0.11 | 0.06 | 0.12 | 0.41 | 0.21 |
| OURS (1-12 (default)) | 80.7 | 91.4 | 69.4 | **99.3** | **90.3** | **85.6** | **52.7** | **81.35** [6] |
| - Tuned / Total (%) | 0.20 | 0.31 | 0.20 | 0.11 | 0.06 | 0.12 | 0.41 | 0.21 |

Table S17: **Fourier prompt depth per-task results on VTAB-1k [78]** *Specialized* for ViT-Base/16 [23] pretrained on supervised ImageNet-21k.

| ViT-Base/16 [23] (85.8M) | VTAB-1k [78] *Specialized* [4] | | | | Mean |
|---|---|---|---|---|---|
| | Patch Camelyon | EuroSAT | Resisc45 | Retinopathy | |
| FULL [92] | 85.1 | 96.4 | 83.1 | 74.3 | 84.72 |
| VPT-SHALLOW [4] | 78.2 | 92.0 | 75.6 | 72.9 | 79.66 [0] |
| - Tuned / Total (%) | 0.01 | 0.05 | 0.09 | 0.01 | 0.04 |
| VPT-DEEP [4] | 81.8 | 96.1 | 83.4 | 68.4 | 82.43 [2] |
| - Tuned / Total (%) | 1.06 | 1.07 | 0.15 | 0.02 | 0.57 |
| OURS (1 3 5 7 9 11) | 82.9 | 95.2 | 81.8 | 75.1 | 83.73 [3] |
| - Tuned / Total (%) | 1.06 | 0.12 | 0.11 | 0.03 | 0.33 |
| OURS (1-6) | **84.0** | 95.0 | 83.6 | 74.7 | 84.34 [3] |
| - Tuned / Total (%) | 1.06 | 0.12 | 0.11 | 0.03 | 0.33 |
| OURS (7-12) | 83.3 | 95.4 | 82.4 | 74.7 | 83.93 [3] |
| - Tuned / Total (%) | 1.06 | 0.12 | 0.11 | 0.03 | 0.33 |
| OURS (1-12 (default)) | 83.5 | **96.5** | **84.4** | **75.4** | **84.93** [4] |
| - Tuned / Total (%) | 1.06 | 0.12 | 0.11 | 0.03 | 0.33 |

## S2.3 Per-task Results of Fourier Prompt Location on VTAB-1k *Natural* and *Specialized*

Table S18: **Fourier prompt location per-task results on VTAB-1k [78]** *Natural* for ViT-Base/16 [23] pretrained on supervised ImageNet-21k.

| ViT-Base/16 [23] (85.8M) | VTAB-1k [78] *Natural* [7] | | | | | | | Mean |
|---|---|---|---|---|---|---|---|---|
| | CIFAR-100 | Caltech101 | DTD | Flowers102 | Pets | SVHN | Sun397 | |
| FULL [92] | 57.6 | 91.0 | 64.6 | 91.6 | 79.9 | 89.8 | 29.1 | 71.95 |
| VPT-SHALLOW [4] | 77.7 | 86.9 | 62.6 | 97.5 | 87.3 | 74.5 | 51.2 | 76.81 [4] |
| - Tuned / Total (%) | 0.18 | 0.10 | 0.04 | 0.27 | 0.08 | 0.19 | 0.36 | 0.17 |
| VPT-DEEP [4] | 78.8 | 90.8(3) | 65.8 | 98.0 | 88.3 | 78.1 | 49.6 | 78.48 [6] |
| - Tuned / Total (%) | 0.20 | 0.20 | 0.15 | 0.10 | 0.04 | 0.54 | 0.41 | 0.23 |
| OURS-Append | 81.0 | **92.4** | **72.2** | 98.4 | 86.7 | 85.6 | 50.8 | 81.02 [6] |
| - Tuned / Total (%) | 0.20 | 0.31 | 0.20 | 0.11 | 0.06 | 0.12 | 0.41 | 0.21 |
| OURS-Random | **81.9** | 91.8 | 66.0 | 98.3 | 89.2 | 71.7 | 51.5 | 78.62 [6] |
| - Tuned / Total (%) | 0.20 | 0.31 | 0.20 | 0.11 | 0.06 | 0.12 | 0.41 | 0.21 |
| OURS-Prepend (default) | 80.7 | 91.4 | 69.4 | **99.3** | **90.3** | 85.6 | **52.7** | **81.35** [6] |
| - Tuned / Total (%) | 0.20 | 0.31 | 0.20 | 0.11 | 0.06 | 0.12 | 0.41 | 0.21 |

Table S19: **Fourier prompt location per-task results on VTAB-1k [78]** *Specialized* for ViT-Base/16 [23] pretrained on supervised ImageNet-21k.

| ViT-Base/16 [23] (85.8M) | VTAB-1k [78] *Specialized* [4] | | | | Mean |
|---|---|---|---|---|---|
| | Patch Camelyon | EuroSAT | Resisc45 | Retinopathy | |
| FULL [92] | 85.1 | 96.4 | 83.1 | 74.3 | 84.72 |
| VPT-SHALLOW [4] | 78.2 | 92.0 | 75.6 | 72.9 | 79.66 [0] |
| - Tuned / Total (%) | 0.01 | 0.05 | 0.09 | 0.01 | 0.04 |
| VPT-DEEP [4] | 81.8 | 96.1 | 83.4 | 68.4 | 82.43 [2] |
| - Tuned / Total (%) | 1.06 | 1.07 | 0.15 | 0.02 | 0.57 |
| OURS-Append | 83.2 | 95.1 | 81.5 | 75.4 | 83.80 [3] |
| - Tuned / Total (%) | 1.06 | 0.12 | 0.11 | 0.03 | 0.33 |
| OURS-Random | 83.2 | 95.1 | 76.2 | 75.4 | 82.47 [3] |
| - Tuned / Total (%) | 1.06 | 0.12 | 0.11 | 0.03 | 0.33 |
| OURS-Prepend (default) | **83.5** | **96.5** | **84.4** | **75.4** | **84.93** [4] |
| - Tuned / Total (%) | 1.06 | 0.12 | 0.11 | 0.03 | 0.33 |

## S2.4 Per-task Results of Fourier Prompt Dimension on VTAB-1k *Natural* and *Specialized*

Table S20: **Fourier prompt dimension per-task results on VTAB-1k [78] *Natural*** for ViT-Base/16 [23] pretrained on supervised ImageNet-21k.

| ViT-Base/16 [23] (85.8M) | VTAB-1k [78] *Natural* [7] | | | | | | | Mean |
|---|---|---|---|---|---|---|---|---|
| | CIFAR-100 | Caltech101 | DTD | Flowers102 | Pets | SVHN | Sun397 | |
| FULL [92] | 57.6 | 91.0 | 64.6 | 91.6 | 79.9 | 89.8 | 29.1 | 71.95 |
| VPT-SHALLOW [4] | 77.7 | 86.9 | 62.6 | 97.5 | 87.3 | 74.5 | 51.2 | 76.81 [4] |
| - Tuned / Total (%) | 0.18 | 0.10 | 0.04 | 0.27 | 0.08 | 0.19 | 0.36 | 0.17 |
| VPT-DEEP [4] | 78.8 | 90.8(3) | 65.8 | 98.0 | 88.3 | 78.1 | 49.6 | 78.48 [6] |
| - Tuned / Total (%) | 0.20 | 0.20 | 0.15 | 0.10 | 0.04 | 0.54 | 0.41 | 0.23 |
| OURS-Sequence length | 79.8 | **91.6** | **70.3** | 98.5 | 89.6 | 84.0 | 52.3 | 80.88 [6] |
| - Tuned / Total (%) | 0.20 | 0.31 | 0.20 | 0.11 | 0.06 | 0.12 | 0.41 | 0.21 |
| OURS-Hidden | 80.5 | 91.5 | 69.9 | 98.5 | 89.5 | 83.5 | 51.9 | 80.74 [6] |
| - Tuned / Total (%) | 0.20 | 0.31 | 0.20 | 0.11 | 0.06 | 0.12 | 0.41 | 0.21 |
| OURS-Both (default) | **80.7** | 91.4 | 69.4 | **99.3** | **90.3** | **85.6** | 52.7 | **81.35** [6] |
| - Tuned / Total (%) | 0.20 | 0.31 | 0.20 | 0.11 | 0.06 | 0.12 | 0.41 | 0.21 |

Table S21: **Fourier prompt dimension per-task results on VTAB-1k [78] *Specialized*** for ViT-Base/16 [23] pretrained on supervised ImageNet-21k.

| ViT-Base/16 [23] (85.8M) | VTAB-1k [78] *Specialized* [4] | | | | Mean |
|---|---|---|---|---|---|
| | Patch Camelyon | EuroSAT | Resisc45 | Retinopathy | |
| FULL [92] | 85.1 | 96.4 | 83.1 | 74.3 | 84.72 |
| VPT-SHALLOW [4] | 78.2 | 92.0 | 75.6 | 72.9 | 79.66 [0] |
| - Tuned / Total (%) | 0.01 | 0.05 | 0.09 | 0.01 | 0.04 |
| VPT-DEEP [4] | 81.8 | 96.1 | 83.4 | 68.4 | 82.43 [2] |
| - Tuned / Total (%) | 1.06 | 1.07 | 0.15 | 0.02 | 0.57 |
| OURS-Sequence length | 81.5 | 95.3 | 82.5 | 75.0 | 83.57 [3] |
| - Tuned / Total (%) | 1.06 | 0.12 | 0.11 | 0.03 | 0.33 |
| OURS-Hidden | 83.3 | 94.7 | 82.8 | 74.6 | 83.87 [3] |
| - Tuned / Total (%) | 1.06 | 0.12 | 0.11 | 0.03 | 0.33 |
| OURS-Both (default) | **83.5** | **96.5** | **84.4** | **75.4** | **84.93** [4] |
| - Tuned / Total (%) | 1.06 | 0.12 | 0.11 | 0.03 | 0.33 |

## S2.5 Sensitivity of Fourier Prompt Percentages and Prompt Lengths

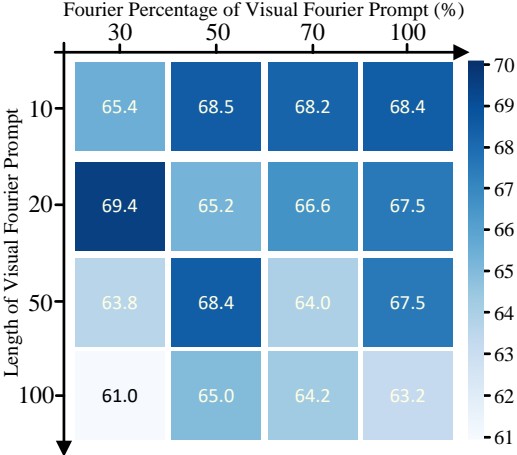

Figure S1: Sensitivity of visual Fourier prompt percentages and its prompt lengths on VTAB-1k [78] DTD.

## S3 Per-task Results on Fourier Percentage

Table S22: **Fourier percentage per-task results on VTAB-1k [78]** *Natural* for ViT-Base/16 [23] pretrained on supervised ImageNet-21k. The highest accuracy among all Fourier percentages are shown in **bold**. Same for Table S23 and S24

| Fourier | VTAB-1k [78] *Natural* [7] | | | | | | |
|---|---|---|---|---|---|---|---|
| Percentage (%) | CIFAR-100 | Caltech101 | DTD | Flowers102 | Pets | SVHN | Sun397 |
| 0 | 78.8 | 90.8 | 65.8 | 97.9 | 88.4 | 76.4 | 49.6 |
| 30 | 79.7 | 91.4 | **69.4** | — | — | 83.1 | 51.3 |
| 50 | 80.3 | **91.4** | 68.5 | **99.3** | **90.3** | **84.3** | **52.7** |
| 70 | **80.7** | 91.3 | 66.6 | — | — | 84.0 | 52.1 |
| 100 | 80.6 | 91.0 | 67.8 | 98.3 | 87.2 | 78.5 | 52.3 |

Table S23: **Fourier percentage per-task results on VTAB-1k [78]** *Specialized* for ViT-Base/16 [23] pretrained on supervised ImageNet-21k.

| Fourier | VTAB-1k [78] *Specialized* (4) | | | |
|---|---|---|---|---|
| Percentage (%) | Patch Camelyon | EuroSAT | Resisc45 | Retinopathy |
| 0 | 82.0 | 96.1 | 83.4 | 68.0 |
| 30 | 82.6 | 95.3 | **84.3** | — |
| 50 | 82.4 | 96.1 | 83.6 | 74.6 |
| 70 | 83.2 | 96.2 | 83.2 | — |
| 100 | **83.3** | **96.3** | 83.1 | **75.4** |

Table S24: **Fourier percentage per-task results on VTAB-1k [78]** *Structured* for ViT-Base/16 [23] pretrained on supervised ImageNet-21k.

| Fourier | VTAB-1k [78] *Structured* [8] | | | | | | | |
|---|---|---|---|---|---|---|---|---|
| Percentage (%) | Clevr/ count | Clevr/ distance | DMLab | KITTI/ distance | dSprites/ location | dSprites/ orientation | SmallNORB/ azimuth | SmallNORB/ elevation |
| 0 | 68.5 | 60.0 | 46.5 | 72.8 | 73.6 | 47.3 | 29.3 | 40.2 |
| 30 | 73.7 | 61.2 | 46.7 | 76.8 | 74.7 | 46.1 | 24.6 | 42.0 |
| 50 | 73.5 | 62.1 | 47.1 | **79.3** | 74.5 | 47.9 | 30.6 | 41.9 |
| 70 | 74.3 | 62.7 | **48.3** | 79.0 | 79.7 | **56.0** | 30.8 | **43.4** |
| 100 | **75.8** | **63.2** | 47.5 | 77.1 | **81.5** | 47.9 | **34.1** | 42.0 |

# S4    Visualization of Attention Map

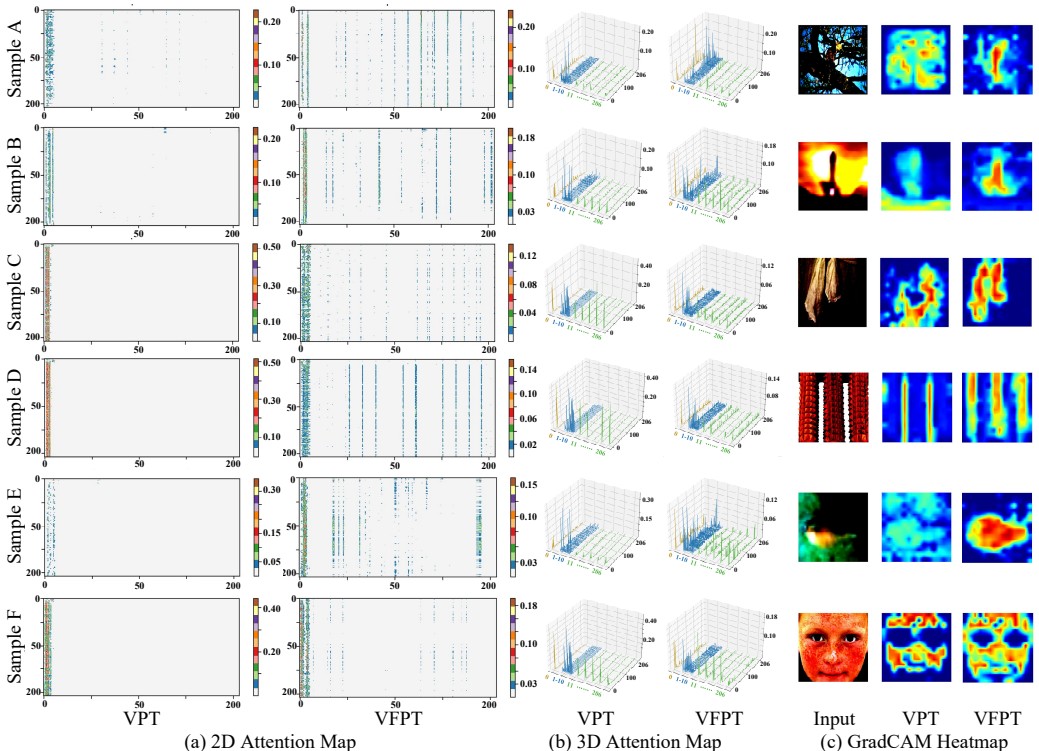

Figure S2: (a) **More visualization results of 2D attention map** on VTAB-1K [78] (b) **Corresponding 3D attention maps**. Figures are best viewed by zooming in. (c) **More visual inspection of VPT and VFPT** using GradCAM [104]. Consistent to our paper, the red regions correspond to high score for class. From left to right are input image after standard data augmentation, GradCAM results for VPT and GradCAM results for VFPT. Figure best viewed in color.

In this section, we present more details and results of visualization of attention maps to support our findings in §4.4. All samples selected from VTAB-1k [78] have the same prompt length (*i.e.*, 10 prompts) with one class token and 196 input patches.

In Fig.S2(a), we can first observe a significant attention distribution in learnable prompts and then a notably higher concentration in global attention scores when integrating visual Fourier prompts, showing consistency with our paper.

In Fig.S2(b), we present more visualization inspection results for VPT and VFPT using Grad-CAM [104]. Overall, we present additional visual evidence to support the notion that the integration of visual Fourier prompts encourage clear foreground-background separation.

# S5 Visualization of Loss Landscape

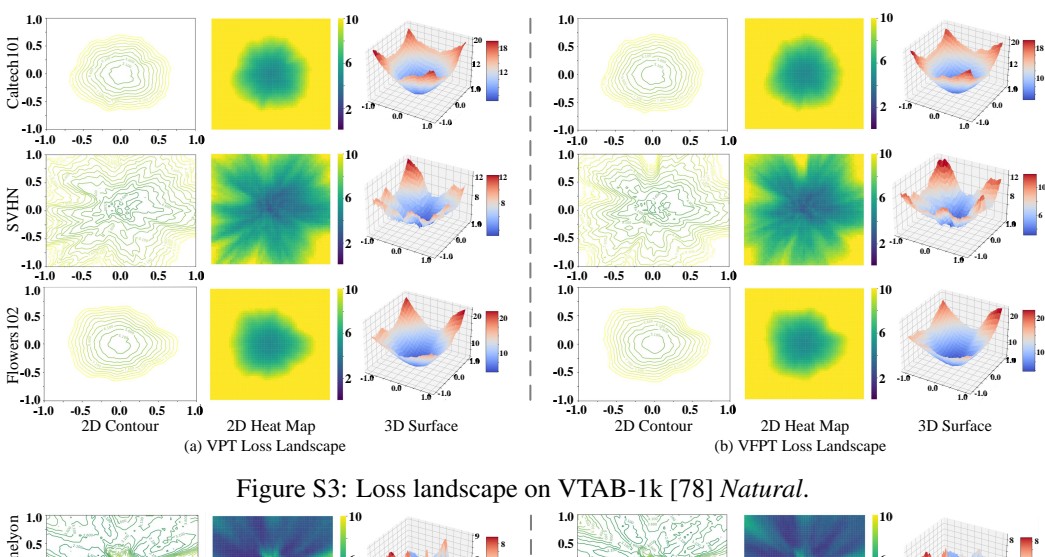

Figure S3: Loss landscape on VTAB-1k [78] *Natural*.

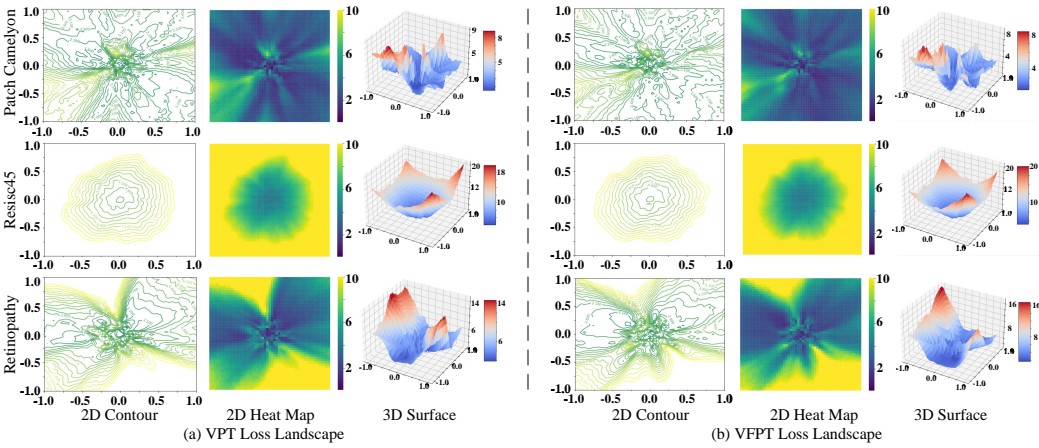

Figure S4: Loss landscape on VTAB-1k [78] *Specialized*.

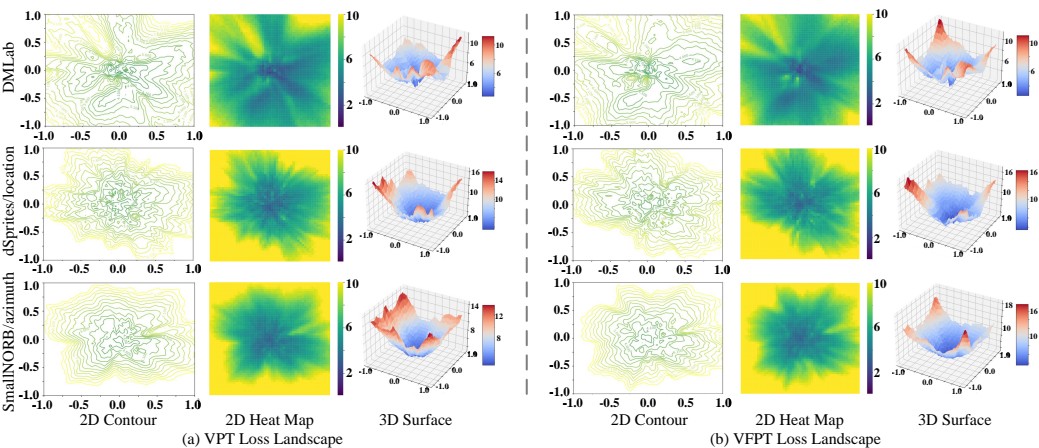

Figure S5: Loss landscape on VTAB-1k [78] *Structured*.

## S6 Extension to Language Tasks

While ViT-Base/16 [23] is structurally similar to BERT [107], we follow [5, 108] and naturally test the efficiency of the VFPT on natural language understanding (NLU) tasks. Specifically, we include BERT-Large [107] for evaluation, and compare full fine-tuning (FULL) [2], Prompt Tuning [2], P-Tuning v2 [108] and $E^2$VPT [5] on SuperGlue [107] dataset: a collection of text classification tasks to test the general language understanding ability. The tasks include natural language inference (RTE and CB), coreference resolution (WSC), sentence completion (COPA), word sense disambiguation (WiC), and question answering (MultiRC (Fla), ReCoRD (F1) and BoolQ). In Table S25, we show that VFPT outperforms FULL and Prompt Tuning and show competitive results to P-Tuning v2 [108]. Considering VFPT is designed for visual-related tasks, and text understanding tasks might not need fruitful frequency domain information, these results are impressive and suggest future work for a general solution across modalities under the *pretrain-then-finetune* paradigm.

Table S25: **Per-task results for SuperGLUE development set [109] with a pretrained BERT-Large [107].** See §S6.

| BERT-Large [107] (335M) | SuperGLUE [107] [8] | | | | | | | | Mean |
|---|---|---|---|---|---|---|---|---|---|
| | BoolQ | CB | COPA | MultiRC (Fla) | ReCoRD (F1) | RTE | WiC | WSC | |
| FULL [2] | 77.7 | 94.6 | 69.0 | 70.5 | 70.6 | 70.4 | 74.9 | 68.3 | 74.50 |
| Prompt Tuning [2] | 67.2 | 80.4 | 55.0 | 59.6 | 44.2 | 53.5 | 63.0 | 64.4 | 60.91 |
| P-Tuning v2 [108] | 73.1 | **94.6** | 73.0 | **70.6** | 72.8 | 78.3 | 75.1 | 68.3 | **75.73** |
| $E^2$VPT [5] | 74.4 | 80.4 | 77.0 | 65.8 | 71.9 | **78.7** | 74.3 | 67.3 | 73.73 |
| OURS | **74.8** | 81.2 | **78.1** | 67.8 | **72.9** | 77.2 | **75.3** | **68.4** | 74.46 |

## S7 Extension of Complexity Analysis

Table S26: **Complexity analysis of fourier percentage settings on CIFAR-100 benchmark.** The percentages in the results indicate the rate of improvement compared to VPT.

| Fourier Percentage (%) | Maximum Memory Consumption (GB) | Training Average Batch Time (s) | Inference Average Batch Time (s) |
|---|---|---|---|
| VPT (0%) | 1.8210 | 0.1140 | 0.0499 |
| VFPT (30%) | 1.8210 (0%) | 0.1169 (+2.54%) | 0.0505 (+1.20%) |
| VFPT (50%) | 1.8210 (0%) | 0.1155 (+1.32%) | 0.0502 (+0.60%) |
| VFPT (70%) | 1.8210 (0%) | 0.1150 (+0.88%) | 0.0500 (+0.20%) |
| VFPT (100%) | 1.8210 (0%) | 0.1150 (+0.88%) | 0.0501 (+0.40%) |

We have provided a detailed comparison of our computational results in this section. More specifically, we experimented with different Fourier percentage settings (*i.e..*, the alpha rate) on the CIFAR-100 benchmark and reported their maximum memory consumption, average training batch time, and average inference batch time. All settings were tested with the same batch size and prompt length. The experiments were conducted on NVIDIA A100-40GB GPUs.

As illustrated in Table S26, no significant increase in maximum memory consumption at the MB level is observed across different Fourier percentage settings. However, we do observe a slight increase in average batch time during both training and inference, on the order of $10^{-3}$ and $10^{-4}$, respectively. This suggests that a lower Fourier percentage incurs a higher computational burden. This effect is likely attributable to suboptimal parallel acceleration and the implementation inefficiencies associated with prompts that have partial Fourier transformation. We will investigate this further in future research.

## S8 Asset License and Consent

The majority of VPT [4] is licensed under CC-BY-NC 4.0. Portions of [4] are available under separate licenses: google-research/task_adaptation and huggingface/transformers are licensed under Apache-2.0; Swin-Transformer [24] and ViT-pytorch [23] are licensed under MIT; and MoCo-v3 [26] and MAE [90] are licensed under CC BY 4.0.

All the datasets included in our study are publicly available (VTAB-1K, FGVC), and all the models are publicly available. We would like to state that the contents in the dataset do NOT represent our views or opinions.

## S9 Reproducibility

VFPT is implemented in Pytorch [91]. Experiments are conducted on NVIDIA A100-40GB GPUs. To guarantee reproducibility, our full implementation shall be publicly released upon paper acceptance. For training schedule, the superior low-complexity of FFT (*i.e.*, $O(n \log n)$) allows for efficient training of visual Fourier prompts with only a slight decrease in training speed (*i.e.*, 2.8% on VTAB-1k [78] compared to VPT).

## S10 Social Impact and Limitations

This study presents VFPT, demonstrating significant and generalizable performance enhancements over state-of-the-art baselines across two benchmarks. The incorporation of the FFT contributes these advantages without necessitating architecture-specific designs or incurring substantial computational overhead under *pretrain-then-finetune* paradigm for large-scale models (see §3). Our approach enjoys advanced model accuracy, and is valuable in real-world computational-sensitive applications, *e.g.*, training machine learning models on edge devices. Moreover, VFPT advances significantly towards achieving generality across datasets, demonstrating substantial performance improvements even when faced with large dataset disparities (see §4). This progress is crucial for the continuous development of PEFT across a wider spectrum of applications.

For potential limitations, drawing inspirations from human visual cognition, our method incorporates spatial and frequency information, which brings an additional hyper-parameter — Fourier percentage (*i.e.*, $\alpha$ in §3.2). However, in practical applications, we observe in §4.2 that dataset disparity (*i.e.*, low disparity tasks prefer small $\alpha$ value, and vice versa) serves as a guideline for selecting an appropriate Fourier percentage. Nonetheless, we argue that the implementation of an automatic Fourier percentage search can further augment efficiency.

## S11 Discussion and Future Work

In §2, we review PEFT methods and the application of the fast Fourier transform in vision. Notably, a recent study [36] in NLP incorporates Fourier transform as a viable PEFT approach, which warrants discussion. Specifically, it learns a set of spectral coefficients of Fourier basis using a LoRA-based approach and then applies the inverse discrete Fourier transform to the spectral matrix, yielding its spatial-domain counterpart as the updated weight change. Although the Fourier basis's orthogonal and expressive advantages reduce the need for extensive parameter fine-tuning, the inverse transform applied to the spectral matrix discards frequency information, ultimately considering only traditional spatial domain features. The parameter-efficient use of the Fourier transform in this study is orthogonal to our method, where both spatial and frequency domain information are integrated (see §3) for enhanced generality (see §4.2) and interpretability (see §4.4).

Despite VFPT systemic effectiveness and simplicity, it also comes with new challenges and unveils some intriguing questions. For example, the balance between spatial and frequency information is presently dictated by task-specific, manually set percentages (see §4.2). Introducing a small network within the VFPT framework to autonomously search for optimal combinations might enhance training efficiency and facilitate additional performance improvements. Another essential future direction deserving of further investigation is the integration of visual information from both the spatial and frequency domains. In §4.5, we demonstrate through ablation studies that integration at the pre-processing stage may not yield satisfactory performance. Consequently, we outline several alternative integration approaches in Table 4, demonstrating that VFPT holds the most advantageous position under the prompt tuning paradigm. Nonetheless, the applicability of this integration to other PEFT methods requires further investigation.

