# OpenReview forum: "Visual Fourier Prompt Tuning"
_NeurIPS.cc/2024/Conference — NeurIPS 2024 poster_

### Official Review · Reviewer_TrZ7 · 2024-07-11

**Soundness:** 3
**Presentation:** 3
**Contribution:** 3
**Rating:** 7
**Confidence:** 5

**Summary:**

This paper introduces Visual Fourier Prompt Tuning (VFPT), a novel approach to parameter-efficient fine-tuning (PEFT) for large-scale Transformer-based vision models. VFPT integrates Fast Fourier Transform (FFT) with prompt tuning, allowing the model to adapt to new tasks effectively by utilizing both spatial and frequency domain information, especially when there is a significant disparity between the datasets used in pre-training and fine-tuning phases. This method not only retains the simplicity and intuitiveness of standard visual prompt tuning but also enhances model performance across various tasks without substantially increasing the parameter count. Empirical results show that VFPT outperforms several state-of-the-art PEFT methods on benchmark datasets like VTAB-1k and FGVC, achieving higher accuracy with fewer parameters. For instance, VFPT uses only 0.66% of the total model parameters while achieving 73.20% mean accuracy on VTAB-1k, surpassing both full fine-tuning and conventional visual prompt tuning methods.

**Strengths:**

1.	The paper is well-structured and clearly written, with each section logically flowing into the next.
2.	VFPT introduces a unique combination of Fast Fourier Transform (FFT) with visual prompt tuning in Transformer-based models, which leverages both spatial and frequency domain information.
3.	The empirical results are robust, demonstrating VFPT's superiority over existing state-of-the-art parameter-efficient fine-tuning methods. The paper includes detailed comparative analyses, showing VFPT's performance advantages across multiple benchmark datasets.
4.	VFPT has significant implications for the scalability and efficiency of Transformer-based models in vision tasks. The ability to maintain high performance with fewer parameters makes large-scale models more accessible and practical for a wider range of applications.
5.	The paper discusses the theoretical implications of integrating FFT within visual prompt tuning, noting improvements in the optimization landscape and model interpretability. VFPT promotes a smoother loss landscape, which correlates with better generalization and lower test error rates.

**Weaknesses:**

1.	The abstract states that the proposed method, VFPT, utilizes only 0.57% of the model parameters to achieve a performance of 73.20% mean accuracy on VTAB-1k dataset. However, Table 1 contradicts this by listing the parameter usage as 0.66%. This discrepancy raises concerns about the accuracy of the reported statistics.
2.	The paper does not fully elucidate the complexity introduced by integrating FFT into the prompt tuning process. While the method is praised for its parameter efficiency, the computational overhead, especially in terms of runtime and memory consumption during the FFT operations, is not thoroughly discussed.
3.	Table 3 reveals that VFPT appears to underperform relative to some established partial tuning and extra module methods when applied to models pretrained with MAE self-supervised objective. This might indicate that VFPT's reliance on Fourier transforms does not synergize as effectively with the feature representations or data distributions typical of self-supervised learning models. Understanding why VFPT performs less effectively in this context is crucial for refining the approach or identifying its optimal application scenarios.

**Questions:**

Table 5(b) explores the effects of different Fourier Prompt Locations (Prepend, Append, and Random) on the performance of Visual Fourier Prompt Tuning (VFPT). The results suggest variations in performance based on the placement of these prompts within the input sequence. Since VPT mentions that different prompt locations are mathematically equivalent, I wonder why the location of Fourier prompts impacts model performance? The following sentence is from section 3.2 of the original VPT paper: “Notably for ViT, $x_N$ is invariant to the location of prompts since they are inserted after positional encoding, e.g., $[\mathbf{x_0}, \mathbf{P}, \mathbf{E_0}]$ and $[\mathbf{x_0}, \mathbf{E_0}, \mathbf{P}]$ are mathematically equivalent. This also applies to VPT-Deep.”

**Limitations:**

The limitations have been discussed in Appendix S9.

---

> ### Author Rebuttal · Authors · 2024-08-07
>
> Dear Reviewer TrZ7,
>
> We sincerely appreciate your time and effort in reviewing our paper and providing valuable comments. We provide explanations to your questions point-by-point in the following.
>
> **Q1: Regarding the discrepancy of tuned parameter rate.**
>
> **A1:** Sorry for the confusion. We want to clarify that the average percentage 0.57% mentioned in the abstract is exclusive to the results obtained on the VTAB-1k benchmark, which comprises 19 subset datasets. While 0.66 % in Table 1 is the average percentage of tuned parameters required by the overall 24 tasks. i.e., encompasses both the VTAB-1k benchmark and the FGVC datasets (i.e., 19 from VTAB-1k and 5 from FGVC). We will revise accordingly to make it more clear.
>
> **Q2: Regarding complexity analysis.**
>
> **A2:** Thank you for the excellent suggestion. We have provided a detailed comparison of our computational results below. More specifically, we experimented with different Fourier percentage settings (i.e., the alpha rate) on the CIFAR-100 benchmark and reported their maximum memory consumption, average training batch time, and average inference batch time. All settings were tested with the same batch size and prompt length. The experiments were conducted on NVIDIA A100-40GB GPUs. We’ll supplement these results in the revision.
>
> |    Configurations   | Maximum Memory Consumption (GB) | Training Average Batch Time (s) | Inference Average Batch Time (s) |
> | ---------------- | ------------------------------ | ------------------------------- | -------------------------------- |
> | VPT (alpha=0.0)  | 1.8210                     | 0.1140                          | 0.0499                           |
> | VFPT (alpha=0.3) | 1.8210 (0%)                | 0.1169 (+2.54%)         | 0.0505 (+1.20%)        |
> | VFPT (alpha=0.5) | 1.8210 (0%)                | 0.1155 (+1.32%)         | 0.0502 (+0.60%)        |
> | VFPT (alpha=0.7) | 1.8210 (0%)                | 0.1150 (+0.88%)    | 0.0500 (+0.20%)             |
> | VFPT (alpha=1.0) | 1.8210 (0%)                | 0.1150 (+0.88%)    | 0.0501 (+0.40%)             |
>
>
>
> **Q3: Regarding the performance in MAE tasks.**
>
> **A3:** Thank you for the insightful question. There are two main reasons why VFPT performs less effectively in MAE objectives.
>
> **First**, our work is built on top of VPT, which has significantly low performance on different pretrained objectives. VPT claims this may be due to the training strategies being fundamentally different between the two self-supervised ViTs and the supervised ones. Though in this work, we manage to significantly narrow this gap (e.g., 53.59% vs. 36.02% under MAE on VTAB-1k Natural) by introducing frequency domain information, the discrepancy in the training objectives might still cause lower performance.
>
> **Second**, we observe an interesting phenomenon that during prompt tuning, the evaluation performance in self-supervised objective tasks is less stable compared to supervised tasks (e.g., 0.33 vs. 1.30 average standard deviation on VTAB-1k Natural). This observation further supports our claim above. In the future, we plan to explore and further mitigate the gap between different pretrained objectives.
>
> **Q4: Regarding Fourier Prompt Locations.**
>
> **A4:** Thank you for the question. We want to clarify that the claim in VPT holds when switching between $E_{0}$ and $P$, as it only changes the position considering **all prompts as an entire block**. In contrast, in VFPT, the Fourier operation is introduced **within prompts**, indicating that it still exists in different location settings when the Fourier percentage is determined. Our ablative study on Fourier Prompt Location investigates how we incorporate Visual Fourier Prompts within the prompt representation. Different locations result in varied prompt representations, thereby distinctly impacting model performance.
>
> We appreciate your thoughtful comments. We hope our response addresses your concerns. Please let us know if there are any additional questions, and we will be happy to discuss further.

---

> > ### Comment · Reviewer_TrZ7 · 2024-08-09
> >
> > I appreciate your detailed response. The authors have addressed all the concerns I raised in my initial review, and after considering the other feedback, I have raised my score to 7.

---

> > > ### Author Response · Authors · 2024-08-09
> > > **Thank you for the prompt reply**
> > >
> > > Thank you for your prompt response and the constructive feedback, which is crucial for enhancing our work.

---

### Official Review · Reviewer_9y4B · 2024-07-12

**Soundness:** 3
**Presentation:** 3
**Contribution:** 3
**Rating:** 6
**Confidence:** 3

**Summary:**

This paper introduces Visual Fourier Prompt Tuning (VFPT), an approach for parameter-efficient fine-tuning of large vision models. VFPT integrates Fast Fourier Transform (FFT) operations into visual prompt tuning, allowing it to incorporate both spatial and frequency domain information. The method demonstrates superior performance and generalizability across various tasks and datasets, particularly when there are significant disparities between pretraining and fine-tuning data. VFPT outperforms several state-of-the-art baselines on benchmarks like VTAB-1k and FGVC, using only a small fraction of trainable parameters. The authors provide extensive empirical results, ablation studies, and visualizations to demonstrate the effectiveness of their approach. They also explore the optimization landscape and interpretability aspects of VFPT.

**Strengths:**

Originality:

The paper presents an approach by integrating Fast Fourier Transform (FFT) operations into visual prompt tuning. This is an original idea that bridges frequency domain analysis with parameter-efficient fine-tuning of large vision models. The authors draw inspiration from human visual cognition to incorporate both spatial and frequency domain information, which is a creative angle not explored in previous visual prompt tuning methods.

Quality:

- Comprehensive experiments are conducted across multiple benchmarks (VTAB-1k, FGVC) and model architectures (ViT, Swin Transformer).

- Rigorous ablation studies examine various components of the proposed method.

- The authors provide in-depth analysis of the optimization landscape and interpretability aspects.

- Error bars and statistical significance are reported for main results.

- The paper includes thorough comparisons with state-of-the-art baselines.

Clarity:

The paper is well-structured and clearly written.

Significance:

- It addresses a key challenge in parameter-efficient fine-tuning - performance degradation when there's a large disparity between pretraining and fine-tuning datasets.

- The method achieves strong performance while using only a small fraction of trainable parameters, which is crucial for adapting large models efficiently.

- The approach is general and can potentially be applied to various vision transformer architectures.

**Weaknesses:**

1. The paper lacks a rigorous theoretical analysis of why integrating Fourier components improves generalization. While empirical results are strong, a deeper theoretical investigation could provide insights into:

- Why frequency domain information helps with dataset disparities

- How the balance between spatial and frequency information affects model behavior

- Potential limitations or failure cases of the approach

Suggestion: Develop a theoretical framework explaining the interplay between spatial and frequency domain information in prompt tuning. This could involve analyzing the properties of the loss landscape or examining how Fourier components affect the model's representation space.

2. The Fourier percentage (α) is a crucial hyperparameter, but its selection process is not fully automated. The paper suggests using dataset disparity as a guideline, but this may not always be easily quantifiable.

Suggestion: Develop an adaptive method to automatically determine the optimal Fourier percentage based on task characteristics. This could involve a small meta-network that predicts the optimal α, or a dynamic adjustment mechanism during training.

3. While the paper mentions a slight decrease in training speed (2.8% on VTAB-1k), a more comprehensive analysis of computational trade-offs is missing. This is important for practitioners considering adopting the method.

Suggestion: Provide a detailed analysis of computational overhead across different settings, including:

- Training time comparisons with baselines

- Memory usage

- Inference time implications

- Potential optimizations to reduce overhead

4. The paper primarily uses FFT as a black-box operation. A deeper exploration of how different frequency components contribute to performance could yield additional insights.

Suggestion: Conduct experiments analyzing the impact of different frequency bands on model performance. This could involve selectively filtering certain frequency components or visualizing which frequencies are most important for different tasks.

5. The generalization to other architectures is not clear. While the method is tested on ViT and Swin Transformer, its applicability to other architecture types (e.g., convolutional networks, MLP-Mixers) is not explored.

Suggestion: Extend the experimentation to a broader range of architecture types to demonstrate the method's generality. If challenges arise, analyze and discuss the limitations of applying VFPT to different model families.

6. The paper doesn't deeply explore how Fourier prompts interact with regular prompts or how they influence different layers of the network.

Suggestion: Conduct a layer-wise analysis of how Fourier prompts affect feature representations throughout the network. Visualize and quantify the interactions between Fourier and regular prompts to provide insights into their complementary roles.

7. The paper doesn't thoroughly address the robustness of VFPT to different initializations or potential instabilities during training.

Suggestions: Perform a comprehensive stability analysis, including:

- Sensitivity to different random seeds

- Learning rate schedules that work well with VFPT

- Potential gradient issues or training instabilities

- Robustness to different prompt lengths and configurations

8. While benchmark performance is strong, the paper lacks discussion on how VFPT performs on real-world, large-scale applications outside standard benchmarks.

Suggestion: Include case studies or experiments on practical, large-scale vision tasks (e.g., autonomous driving, medical imaging) to demonstrate the method's effectiveness in real-world scenarios. Discuss any challenges or modifications needed for such applications.

**Questions:**

Please refer to the weaknesses section.

**Limitations:**

Limitations:

The authors do discuss limitations in Section S9 of the appendix, which is a positive point. They specifically mention the introduction of an additional hyperparameter (Fourier percentage) and acknowledge that while dataset disparity can guide its selection, an automatic search method could further improve efficiency. This shows awareness of a key limitation in their approach.

However, the limitations discussion could be more comprehensive. For example:

1. They could elaborate on potential scenarios where VFPT might not perform well.

2. Discussion of scalability limitations or computational constraints for very large models is not clearly addressed.

3. Potential limitations in applying VFPT to other domains or model architectures could be explored further.

Societal Impact:

The authors do discuss social impact in Section S9 of the appendix, which is commendable. They highlight positive aspects such as:

1. Potential for improving model accuracy without substantial computational overhead.

2. Applicability in computationally-sensitive real-world applications.

3. Advancements towards achieving generality across datasets.

However, the discussion of potential negative societal impacts is limited. The authors could have explored:

1. Potential misuse of more efficient and generalizable models in privacy-invading or surveillance applications.

2. Environmental impacts of encouraging more fine-tuning experiments.

3. Potential biases that might be introduced or amplified through this method.

---

> ### Author Rebuttal · Authors · 2024-08-07
>
> Dear Reviewer 9y4B,
>
> We sincerely appreciate your time and effort in reviewing our paper and providing detailed comments and suggestions, which are crucial for improving our work. We provide explanations to your questions point-by-point in the following.
>
> **Q1: Regarding theoretical analysis.**
>
> **A1:** This is indeed an excellent suggestion. While we empirically investigate why VFPT achieves better performance and generalizes well across various tasks from an optimization perspective, we also want to highlight several potential directions for future theoretical analysis:
> - The influence of the self-attention mechanism: the self-attention mechanism is the key component of transformers, we plan to conduct a deeper exploration into how the incorporation of frequency information influences the attention module.
> - The potential mitigation of low-pass filter phenomena [1] in transformers. The work in [1] indicates that as ViT scales up, excessive low-pass filtering can cause attention collapse. In future research, we aim to theoretically analyze whether the Fourier transform can reduce low-frequency passing and introduce high-frequency components to mitigate this issue.
>
> [1] Anti-Oversmoothing in Deep Vision Transformers via the Fourier Domain Analysis: From Theory to Practice. ICLR 2022.
>
> **Q2: Automatically determine the optimal Fourier percentage.**
>
> **A2:** We are exploring introducing a small network within the VFPT framework to autonomously search for optimal combinations as mentioned in Appendix S10 (line 733-735). This approach might enhance training efficiency and facilitate additional performance improvements. We plan to conduct more experiments in this direction. Thank you for the suggestion.
>
> **Q3: A more comprehensive analysis of computational trade-offs.**
>
> **A3:** Following the suggestion, we provide a detailed computational analysis with different settings below. Specifically, we experiment with different Fourier percentages on the CIFAR-100 benchmark and report their maximum memory consumption, average training batch time, and average inference batch time. All settings are tested with the same batch size and prompt length. The experiments are conducted on NVIDIA A100-40GB GPUs.
>
>
> |   Settings    | Maximum Memory Consumption (GB) | Training Average Batch Time (s) | Inference Average Batch Time (s) |
> | ---------------- | ------------------------------ | ------------------------------- | -------------------------------- |
> | VPT (alpha=0.0)  | 1.8210                     | 0.1140                          | 0.0499                           |
> | VFPT (alpha=0.3) | 1.8210 (0%)                | 0.1169 (+2.54%)          | 0.0505 (+1.20%)       |
> | VFPT (alpha=0.5) | 1.8210 (0%)                | 0.1155 (+1.32%)         | 0.0502 (+0.60%)        |
> | VFPT (alpha=0.7) | 1.8210 (0%)                | 0.1150 (+0.88%)       | 0.0500 (+0.20%)          |
> | VFPT (alpha=1.0) | 1.8210 (0%)                | 0.1150 (+0.88%)    | 0.0501 (+0.40%)             |
>
>
> **Q4: Analyzing the impact of different frequency bands.**
>
> **A4:** Thank you for the insightful suggestion. Following the suggestion, we further visualize the spectral response of an attention map in the last layer (**Figure 1 in the rebuttal PDF**) on the KITTI/distance benchmark following Anti-Oversmoothing [1]. The preliminary results show that VFPT effectively filters out specific low-frequency components while introducing high-frequency components when compared to VPT. This observation suggests that VFPT may effectively mitigate the attention collapse phenomenon (i.e., as the transformer goes deeper, the attention maps gradually become similar and even much the same after certain layers), as introduced in [1-3]. This observation is consistent with VFPT’s superior performance when compared to VPT. We’ll supplement the analysis with further discussion in the revision.
>
> [2] Improving vision transformers by revisiting high-frequency components. ECCV 2022.
>
> [3] Deepvit: Towards deeper vision transformer. arXiv 2021
>
> (Due to character limitations, we will continue to answer in another comment.)

---

> ### Author Response · Authors · 2024-08-07
>
> (Continue from where rebuttal ends.)
>
> **Q5: The generalization to other model families.**
>
> **A5:** Thank you for the great suggestion. Following VPT, we further conduct experiments on ConvNeXt-Base pretrained on ImageNet21K. ConvNeXt-Base is a convolutional neural network with size 87.6M. We follow the padding strategy to incorporate learnable prompts for the input images for an intuitive setup. The results presented below show a noticeable improvement, indicating the generalization of VFPT to other architectures.
>
> |  ConvNeXt-Base    | VTAB-1k Specialized (4) |
> | ---- | ------------------- |
> | FULL FT | 83.73%              |
> | VPT  | 83.00%              |
> | VFPT | 83.69%              |
>
> In addition, we conducted a preliminary study on a vision-language model: CLIP. Specifically, we applied VFPT to the vision encoder of CLIP using the optimal settings from this work. The results on ImageNet are shown in the table below. As seen, there is a noticeable improvement compared to the standard CLIP, indicating the generalization of VFPT to different models. We’ll provide these additional results and discussions in the revision.
>
> |    ImageNet  | Tuned / Total (%) | Accuracy (%) |
> | -------------- | ----------------- | ------------ |
> | CLIP (zero-shot) | 0.00              | 66.73       |
> | CLIP + VPT     | 0.22              | 70.30        |
> | CLIP + VFPT    | 0.21              | 71.49        |
>
> **Q6: Regarding the interaction of Fourier prompts and regular prompts.**
>
> **A6:** We conduct experiments analyzing the impact of prompts, as detailed in Section 4.4. We observe from the 3D attention map (Figure 4(a)) that there is a pronounced accumulation of attention scores at both Fourier and regular prompt locations, indicating that these prompts substantially impact the feature representation learning. These observations are from the raw average attention head in the last layer of VFPT, and we plan to investigate how they influence different layers of the network as suggested.
>
> **Q7: Regarding model robustness and stability.**
>
> **A7:** Thank you for the great suggestion. In our experiments, all results are averaged over three runs with different random seeds to mitigate potential sensitivity issues related to random seed variation. Additionally, we conduct the same grid search using the same learning rate and weight decay as VPT and E2VPT, detailed in Section 4.1. We further include an ablation study on different prompt lengths and configurations in Appendix 2.5 to evaluate the robustness across varying prompt lengths and Fourier percentages.
>
> Following the suggestion, we conduct an additional ablation study with different initialization techniques (i.e., xavier uniform [4] and He. normal [5] initialization scheme). As seen, both He and xavier initialization show competitive results, validating the robustness of VFPT w.r.t. prompt initialization. We’ll supplement the results with further discussion in the revision.
>
>
>
> |    Initialization     | VTAB-1k Specialized (4) |
> | --------------------- | ----------------------- |
> | VFPT-He.              | 84.88%                  |
> | VFPT-Xavier (default) | 84.93%                  |
>
> [4] Understanding the difficulty of training deep feedforward neural networks. AISTATS 2010.
>
> [5] Delving deep into rectifiers: Surpassing human-level performance on imagenet classification. ICCV 2015.
>
>
> **Q8: How does VFPT perform on real-world, large-scale applications.**
>
> **A8:** Thank you for the question. We want to highlight that VTAB-1k includes several insightful scenes valuable for real-world and diverse applications. For instance, the **specialized group** contains two sub-groups: remote sensing and medical. These are particularly useful for applications like satellite monitoring and medical diagnosis. Many tasks from the **structured group** are generated from simulated environments, contributing to object detection in virtual video game contexts. Our results on VTAB-1k demonstrate the potential of VFPT for various applications. In the future, we plan to employ VFPT to tackle more challenging real-world applications, such as out-of-distribution detection.
>
> **Q9: Regarding Limitations and Social Impact.**
>
> **A9:** Thank you for all the excellent suggestions. We’ll provide more discussion according to the suggestion in our revision.
>
> We appreciate your thoughtful comments. We hope our response addresses your concerns. Please let us know if there are any additional questions, and we will be happy to discuss further.

---

> > ### Comment · Reviewer_9y4B · 2024-08-12
> > **Acknowledging response**
> >
> > The authors have addressed my comments. Considering the feedback from the other reviewers and the authors' responses, I raise my rating to a weak accept for this submission.

---

> > > ### Author Response · Authors · 2024-08-12
> > > **Thank you for your positive response**
> > >
> > > We are delighted to hear that we have successfully addressed all your concerns and received your approval of the work.

---

### Official Review · Reviewer_PyKu · 2024-07-18

**Soundness:** 3
**Presentation:** 3
**Contribution:** 3
**Rating:** 6
**Confidence:** 4

**Summary:**

This paper proposes Visual Fourier Prompt Tuning (VFPT) to address performance degradation in parameter-efficient finetuning (PEFT) methods caused by dataset disparities. VFPT integrates Fast Fourier Transform (FFT) into prompt embeddings, enhancing performance with minimal parameter usage. This work is built upon the VPT method. The key innovation is to integrate frequency domain information using FFT to VPT. The performance is evaluated on VTAB-1k and FGVC.

**Strengths:**

The experiments are comprehensive and performance on several benchmark are impressive as well. Study of Interpretability shows some interesting findings can be helpful for future research. The ablation study is completed. We can see compared with VPT baseline, FFT’s effectiveness has been well demonstrated.
The demonstration including figure/tables of this paper is very well.

**Weaknesses:**

Overall, this is a well-organized paper.  I have several questions regarding the methods and results;
1. What is the larger model’s performance when applying the VFPT? Whether the VFPT can work well with large models is an important question that hasn't been answered in the current version. As the model size grows, efficient, prompt learning can be more critical. However, current results are mainly on a relatively small size of models.
2. Whether VFPT can be integrated into the CLIP model to enhance zero-shot performance is also an interesting question. I am very interested in vision and text encoders and whether VFPT can be applied to CLIP models.
3. Current results do not answer the first question in the introduction: “Can prompt tuning generalize across datasets with varying disparities?”. It confuses me because I think of the transferable prompt design, which is trained on one dataset and then transferred to another. This motivation, to me, needs a further rephrase or validation.

**Questions:**

I have several expectations for seeing the potential of VFPT. If the author can address these questions, I will change my scores.

**Limitations:**

Yes, the authors discussed the limitation of their methods.

---

> ### Author Rebuttal · Authors · 2024-08-07
>
> Dear Reviewer PyKu,
>
> We sincerely appreciate your time and effort in reviewing our paper and providing valuable comments. We provide explanations to your questions point-by-point in the following.
>
> **Q1: Regarding the VFPT performance with larger models.**
>
> **A1:** Thank you for the great suggestion. We use ViT-Base and Swin-Base for comparison as they are the most commonly adopted models in previous works. We completely agree that prompt learning is also important for large models. Based on your suggestion, we have conducted additional experiments by applying VFPT to the ViT-Huge (632M), which is significantly larger than the ViT-Base (86M) and Swin-Base (86.7M). The preliminary results in the table below show that VFPT consistently achieves better performance with larger models. We will include the full results with further discussion in the revision. Thank you again for your suggestion.
>
> | ViT-Huge  | Natural (7) | Specialized (4) | Structured (8) |
> | ---- | ----------- | --------------- | -------------- |
> | VPT  | 77.9        | 83.3            | 52.2           |
> | VFPT | 78.9        | 84.4            | 57.0           |
>
> **Q2: Regarding the integration of VFPT into the CLIP model.**
>
> **A2:** Thank you for the insightful question. The proposed VFPT is a general and robust solution that can be applied to different models. Based on your suggestion, we conducted a preliminary study on CLIP. Specifically, we applied VFPT to the vision encoder of CLIP using the optimal settings from this work. The results on ImageNet are shown in the table below. As seen, there is a noticeable improvement compared to the standard CLIP, indicating the flexibility of adapting VFPT to different models. We appreciate your suggestion and plan to investigate further in the direction of prompt tuning on vision-language models.
>
> |    ImageNet  | Tuned / Total (%) | Accuracy (%) |
> | -------------- | ----------------- | ------------ |
> | CLIP (zero-shot) | 0.00              | 66.73       |
> | CLIP + VPT     | 0.22              | 70.30        |
> | CLIP + VFPT    | 0.21              | 71.49        |
>
> **Q3: Regarding the motivation of ‘Can prompt tuning generalize across datasets with varying disparities?**
>
> **A3:** Sorry for the confusion. We would like to provide some clarification. **First**, this question is motivated by the observation that although original prompt tuning methods (e.g., VPT) have achieved promising results, significant performance degradation occurs when there is a substantial disparity between the datasets used in the pretraining and finetuning phases. The purpose of designing VFPT is to bridge this gap by incorporating frequency domain information. **Second**, we try to answer this question in Section 4.2 (lines 240-268). The empirical results indicate that VFPT achieves noticeably better performance on datasets with large disparities compared to other baselines. We’ll rephrase the question during revision to enhance the clarity.
>
> We appreciate your thoughtful comments. We hope our response addresses your concerns. Please let us know if there are any additional questions, and we will be happy to discuss further.

---

> > ### Comment · Reviewer_PyKu · 2024-08-12
> > **Thanks for the rebuttal**
> >
> > The improvement over scaling up of models and CLIP is not marginal; hence, it enhances the authors' claim.

---

> > > ### Author Response · Authors · 2024-08-13
> > > **Thank you for your valuable suggestions**
> > >
> > > Thank you for your valuable suggestions. We are immensely appreciative of the discussions on the performance of large models and vision-language models, as they clearly illuminate the future direction of our work.

---

### Author Rebuttal · Authors · 2024-08-07

To All Reviewers:

We sincerely thank all reviewers for your valuable suggestions and constructive feedback. We have revised our paper accordingly. The major changes are as follows:
- We’ve conducted additional experiments on the large model (i.e., ViT-Huge), vision-language model (i.e., CLIP), and other architectures (i.e., ConvNeXt-Base), following Reviewer PyKu and 9y4B's suggestion.
- We’ve added additional discussions with real-world, large-scale applications, as suggested by Reviewer 9y4B.
- We’ve rephrased our exploration and theoretical analysis of VFPT, as suggested by Reviewer 9y4B.
- We’ve tried to visualize the spectral response of an attention map in the last layer in Figure 1 of the rebuttal PDF, as suggested by Reviewer 9y4B.
- We’ve provided details about the robustness of VFPT and discussions of determining Fourier percentage automatically, as suggested by Reviewer 9y4B.
- We’ve supplemented runtime analysis and comparison, as suggested by Reviewer 9y4B and TrZ7.
- We’ve provided additional discussion on the less effective performance in self-supervised tasks, as pointed out by Reviewer TrZ7.

We hope our response addresses the concerns from the reviewers. Please let us know if there are any additional questions, and we will be happy to discuss further.

Best Regards,\
Authors

---

### Author Response · Authors · 2024-08-12
**Looking forward to the discussion**

Dear Reviewers,

We sincerely appreciate the time and effort you've devoted to reviewing our work. We understand that your schedule may be quite busy, especially during the weekend. As the authors-reviewer discussion phase draws to a close, we kindly request your attention to our responses (special thanks to Reviewer TrZ7 for acknowledging our response). Our aim is to gain insights into whether our responses effectively address your concerns, and to ascertain if there are any additional questions or points you would like to discuss.

We look forward to the opportunity for further discussion with you. Thank you for your thoughtful consideration.

Best regards,\
The Authors

---

### Decision · Program_Chairs · 2024-09-25

**Decision:**

Accept (poster)

**Comment:**

The paper proposes a novel parameter-efficient finetuning method for vision models that relies on the fast Fourier transform. This allows for strong performance while saving in number of adapted parameters.

This paper has received A,A,WA ratings and the rebuttal has addressed the concerns and questions raised by the reviewers.